# Caspase-2 promotes AMPA receptor internalization and cognitive flexibility via mTORC2-AKT-GSK3β signaling

Zhi-Xiang Xu[1], Ji-Wei Tan[1], Haifei Xu[1], Cassandra J. Hill[1], Olga Ostrovskaya[1], Kirill A. Martemyanov [1] & Baoji Xu [1]

Caspase-2 is the most evolutionarily conserved member in the caspase family of proteases and is constitutively expressed in most cell types including neurons; however, its physiological function remains largely unknown. Here we report that caspase-2 plays a critical role in synaptic plasticity and cognitive flexibility. We found that caspase-2 deficiency led to deficits in dendritic spine pruning, internalization of AMPA receptors and long-term depression. Our results indicate that caspase-2 degrades Rictor, a key mTOR complex 2 (mTORC2) component, to inhibit Akt activation, which leads to enhancement of the GSK3β activity and thereby long-term depression. Furthermore, we found that mice lacking caspase-2 displayed elevated levels of anxiety, impairment in reversal water maze learning, and little memory loss over time. These results not only uncover a caspase-2–mTORC2–Akt–GSK3β signaling pathway, but also suggest that caspase-2 is important for memory erasing and normal behaviors by regulating synaptic number and transmission.

---

[1] Department of Neuroscience, The Scripps Research Institute Florida, Jupiter, FL 33458, USA. Correspondence and requests for materials should be addressed to B.X. (email: bxu@scripps.edu)

In many cortical areas of humans and other mammals, synapse numbers increase over a short period in early postnatal life, followed by an extended period when synapse numbers are reduced to reach mature levels[1,2]. Because dendritic spines are the postsynaptic sites for the vast majority of excitatory synapses[3], synapse pruning reflects a decline in dendritic spine density[4–6]. Pruning of dendritic spines depends on neuronal activity and is required for refinement of neuronal connections in the developing brain[6–9]. Deficits in synapse pruning have been associated with mental disorders, such as autism spectrum disorder[10] and schizophrenia[11].

Microglia participate in synapse pruning by devouring eliminated synapses in a complement-dependent manner[12,13]. However, intracellular pathways that tag synapses for elimination remain unclear. Several molecules, including major histocompatibility class I molecules and the precursor of brain-derived neurotrophic factor, have been found to be essential for both synapse pruning and long-term depression (LTD), which refers to enduring decrease of synaptic strength[14–19]. Therefore, it has been proposed that LTD is the electrophysiological manifestation of synapse pruning[20]. Alternatively, LTD and synapse pruning may share some biochemical mechanisms.

Long-term potentiation (LTP) refers to enduring increase in synaptic strength and is the opposing process to LTD. This bidirectional synaptic plasticity is believed to be critically important in enabling the brain to store vast amounts of information[21,22]. A major mechanism for LTD is removal of the α-amino-3-hydroxy-5-methylisozazole-4-propionic acid receptor (AMPAR) from synapses through internalization[23]. One of the key regulators of AMPAR internalization is glycogen synthase kinase 3β (GSK3β), which is required for N-methyl-D-aspartate (NMDA) receptor-dependent LTD (NMDAR-LTD) at the hippocampal Schaffer collateral-CA1 synapses[24]. GSK3β has a high basal activity, which is modulated by the phosphorylation status of Ser9. Dephosphorylation of this residue by Ser/Thr protein phosphatases causes further activation of GSK3β, whereas phosphorylation of Ser9 by a variety of kinases leads to inhibition of its activity. It has been found that the activity of GSK3β is increased during LTD via activation of phosphatase PP1. Conversely, GSK3β inhibitors or activation of Akt, which inhibits the activity of GSK3β by phosphorylating Ser9, blocks the induction of LTD[24]. Many stimuli, such as growth factors and LTP-inducing stimulation, activate Akt through phosphoinositide 3-kinase (PI3K)[24,25]. Furthermore, the mTOR complex 2 (mTORC2) can fully activate Akt via phosphorylation of Ser473[26]. In addition to NMDAR-LTD, the hippocampus displays a form of LTD that is dependent on the activation of metabotropic glutamate receptors, termed mGluR-LTD. The mTORC2 activity is necessary for late-phase LTP[27] and hippocampal mGluR-dependent LTD[28]. However, it is unclear which upstream signaling pathway regulates the activity of mTORC2 to mediate expression of both LTP and LTD.

Caspases are a family of cysteine proteases that are essential for removing excess cells during development via apoptosis[29]. They could also be involved in eliminating excess synapses during development. Indeed, studies suggest that caspase-3 is essential for internalization of AMPARs, LTD, and spine pruning via cleavage of Akt, thus increasing the activity of GSK3β[30–32]. We focused on the role of caspase-2, encoded by the Casp2 gene, in pruning of dendritic spines, internalization of AMPARs, LTD, and behaviors. Caspase-2, also known as NEDD-2[33] or ICH-1[34], is the most evolutionarily conserved member of the caspase family[35]. In the nervous system, caspase-2 has been implicated in neuronal apoptosis induced by several stimuli, including neurotrophic deprivation[36]. Caspase-2 is reported to mediate synaptic and memory deficits in transgenic mice expressing disease-linked variants of amyloid precursor protein[37], tau[38] and huntingtin[39]. Improvement in memory has been observed in disease mouse models when caspase-2 is removed. Nonetheless, the normal functions of caspase-2 in the nervous system have remained largely unknown, due to a lack of any prominent developmental defects in Casp2 KO mice[40].

In this work, we have found that caspase-2 inhibits mTORC2-induced Akt activation by cleaving the mTORC2 scaffold protein Rictor (rapamycin-insensitive companion of mTOR), thereby stimulating the activity of GSK3β and promoting LTD. Our analysis of cultured neurons and mice reveals that caspase-2 deficiency leads to deficits in internalization of AMPARs, impaired LTD expression, reduced spine pruning, elevated anxiety, enhanced memory, and cognitive inflexibility. These findings not only establish a critical role for caspase-2 in synaptic plasticity and cognition, but also uncover a caspase-2–mTORC2–Akt–GSK3β signaling cascade.

## Results

**Caspase-2 decreases spine density by promoting spine pruning.** To test whether caspase-2 plays a role in synapse pruning, we applied the inhibitor Z-VDVAD-FMK to primary hippocampal neuronal cultures that had been in vitro for 21 days (DIV21), then examined the density and diameter of dendritic spines on DIV28. Compared with DMSO (dimethyl sulfoxide; vehicle) -treated neurons, inhibition of caspase-2 significantly increased spine density (Fig. 1a). We further tested the role of caspase-2 in regulation of spine density by using a short-hairpin RNA (shRNA) that was able to efficiently knock down caspase-2 expression in neurons (Supplementary Fig. 1a, b). Compared with neurons expressing control shRNA, spine density increased by 37% in neurons expressing Casp2 shRNA (Fig. 1b). Concurrently, knocking down caspase-2 with shRNA or inhibiting caspase-2 activity with Z-VDVAD-FMK reduced spine head diameter but increased spine length (Supplementary Fig. 1c–f). The effect on spine density relies on the long-form caspase-2 (caspase-2L), as introduction of shRNA-resistant long-form (R2L) but not short-form (R2S) caspase-2 rescued knockdown of endogenous caspase 2 (Supplementary Fig. 1g). This rescue result also demonstrates the specificity of the Casp2 shRNA. In agreement with these in vitro observations, adult Casp2 knockout (KO) mice showed a significantly higher spine density in distal dendrites of CA1 pyramidal neurons compared with their WT littermates (Fig. 1c and supplementary Fig. 1h). These results indicate that the activity of caspase-2 is important for structural plasticity of synapses.

To determine if caspase-2 modulates spine density by increasing spine elimination or reducing spine formation, we imaged dynamics of dendritic spines in cultured neurons on DIV21-28 over 2 h to measure rates of spine formation and spine elimination (Fig. 1d). We detected a significant reduction in the spine elimination rate but not in the spine formation rate in neurons expressing Casp2 shRNA, compared with neurons expressing control shRNA (Fig. 1e and Supplementary Fig. 1i). Therefore, caspase-2 decreases spine density by promoting spine pruning.

**Neuronal activity alters subcellular caspase-2 distribution.** Previous studies show that caspase-2 is located in Golgi complexes, mitochondria, nucleus, and dendritic spines[37,41–43]. Indeed, we transfected a construct expressing caspase-2L with a HA-tag at the N terminus into cultured hippocampal neurons and found a large amount of caspase-2L localized to the nucleus and a small amount to dendrites (Fig. 2a). As spine pruning is dependent on sensory experience[6], we examined whether caspase-2 is localized in synapses. We prepared synaptosomes

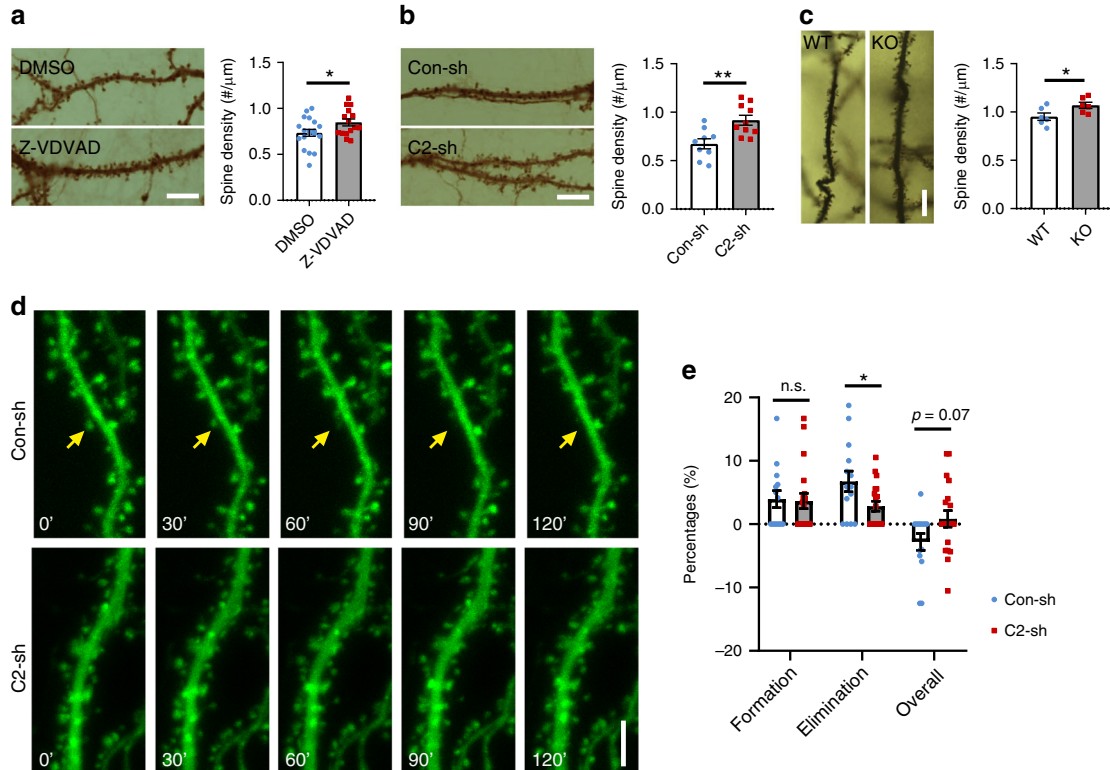

**Fig. 1** Caspase-2 deficiency increases spine density both in vitro and in vivo. **a** Caspase-2 inhibitor (Z-VDVAD) significantly increased spine density in cultured rat hippocampal neurons compared with vehicle DMSO. DMSO, $n = 19$ neurons; Z-VDVAD, $n = 15$ neurons; Scale bar, 10 μm. **b** Knockdown of caspase-2 with shRNA (C2-sh) increased spine density in cultured rat hippocampal neurons compared with control shRNA (Con-sh). Con-sh, $n = 9$ neurons; C2-sh, $n = 10$ neurons; Scale bar, 10 μm. **c** Increased spine density in CA1 neurons of *Casp2* KO mice. WT and KO, $n = 6$ mice; Scale bar, 10 μm. **d**, **e** Live-cell imaging revealed that spine elimination, but not spine formation, was impaired in caspase-2 knockdown neurons. Con-sh, $n = 14$ neurons; C2-sh, $n = 19$ neurons; Scale bar, 5 μm. Neurons were from at least three independent rat hippocampal cultures for analysis of dendritic spines. Data are expressed as mean ± SEM. Two-tailed Student's *t* test: *$p < 0.05$ and **$p < 0.01$. Source data are provided as a Source Data file

from mouse hippocampi. In agreement with a previous report[37], immunoblotting analysis revealed that caspase-2 was indeed present in synaptosomes (Fig. 2b). We next tested if neuronal activity enhances synaptic localization of caspase-2 by treating cultured cortical neurons with 40 μM NMDA for 5 min. Brief NMDA treatment did not change the gene expression of caspase-2 (Supplementary Fig. 2a). However, the amount of caspase-2 in the nucleus was significantly reduced, while its cytosolic portion was increased after elevated neuronal activity (Fig. 2c and Supplementary Fig. 2b). To our surprise, the level of synaptosomal caspase-2 did not increase after NMDA treatment (Fig. 2c). These results suggest that caspase-2 promotes spine pruning by modulating the activity of a signaling cascade outside synapses.

Unlike many other caspases, procaspase-2 is partially active following dimerization[44]. The increased level of cytosolic caspase-2 suggests that neuronal activity might elevate the dimerization and thus the activity of caspase-2 in dendritic shafts, which are part of the cytoplasm. To test this, we employed the bimolecular fluorescence complementation (BiFC) system to examine caspase-2 dimerization upon NMDA stimulation in primary neuronal cultures. BiFC fragments, composed of the caspase-2 prodomain (residues 1–169; known to be sufficient for dimerization) C-terminally fused with VN173 or VC155, have been used to study the dimerization-induced activation of caspase-2 in cultured nonneuronal cells[45]. We found that NMDA treatment increased BiFC signal intensity in dendrites without significantly altering its somal signal (Fig. 2d), indicating that caspase-2 became dimerized and activated upon elevated neuronal activity.

We notice that the BiFC signal is mostly detected in the cytoplasm even in vehicle-treated neurons (Fig. 2d), while endogenous caspase-2 localizes predominantly to the nucleus (Fig. 2a, c). We further used an inhibitor reporter, VDVAD-FITC, to monitor the endogenous caspase-2 activity under different conditions. The inhibitor binds to active caspase-2 and is not cleavable, so that levels of FITC fluorescence are indicative of caspase-2 activity[46]. Consistent to the BiFC assay, brief NMDA treatment did significantly enhance caspase-2 activity in dendrites without altering caspase-2 activity in the soma (Fig. 2e). The activation of caspase-2 in dendrites is unlikely due to apoptosis, because the number of neurons that were positive for cleaved caspase-3 was comparable in vehicle- and NMDA-treated cultures (Supplementary Fig. 2c).

Taken together, these results indicate that activation of NMDA receptors enhances caspase-2 activity in dendrites, in part by mobilizing the protein from the nucleus to the cytoplasm.

**Caspase-2 deficiency impairs hippocampal LTD**. As caspase-2 is present in dendrites and synaptosomes and is essential for spine pruning, we sought to investigate if caspase-2 is functionally involved in synaptic transmission and plasticity. We are particularly interested in LTD, which has been proposed to be the electrophysiological mechanism for synapse elimination[20]. A brief NMDA treatment (40 μM for 5 min) has been used to induce chemical LTD and spine shrinkage in cultural hippocampal neurons[47,48]. We first determined if caspase-2 is required for LTD-induced spine shrinkage. As expected, NMDA treatment

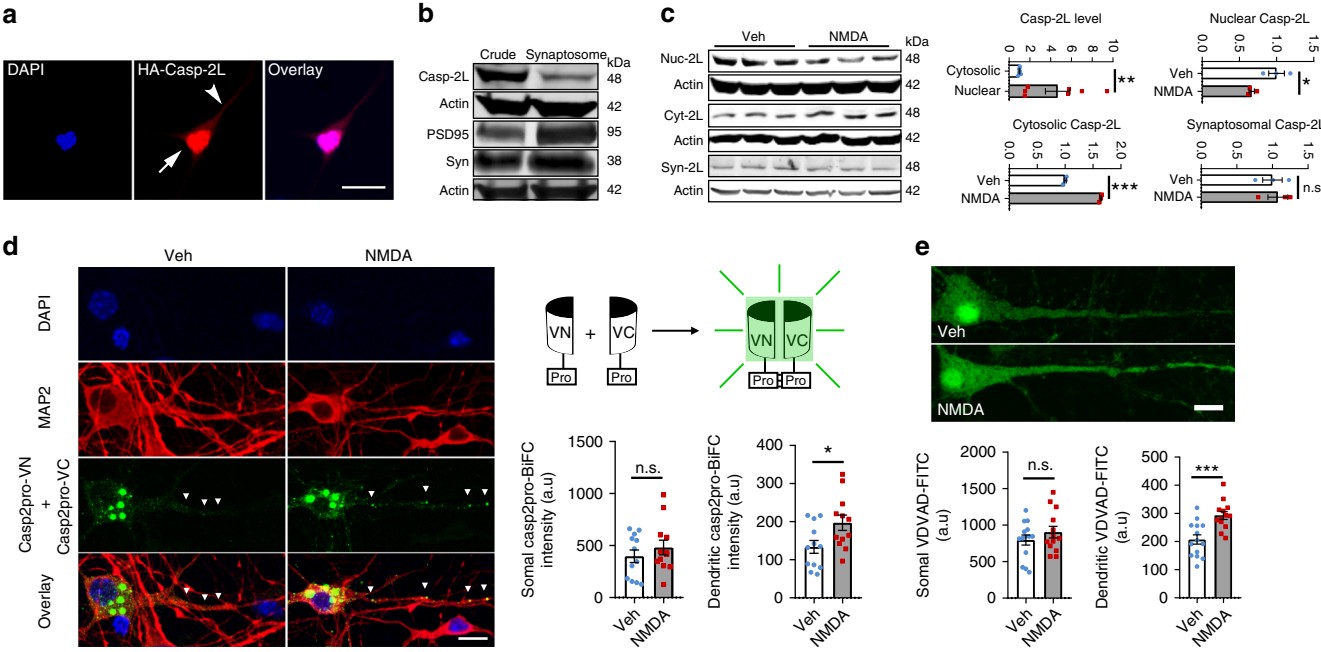

**Fig. 2** Neuronal activity alters subcellular localization of caspase-2. **a** Localization of caspase-2 to the nucleus and dendrites in neurons. Cultured rat hippocampal neurons were transfected with a construct expressing long-form caspase-2 tagged with HA at N terminus (HA-Casp-2L). The arrow and arrow head denote caspase-2 signals in the nucleus and dendrite, respectively. Scale bar, 10 μm. **b** Long-form caspase-2 (Casp-2L) is present in synaptosomes isolated from adult mouse brains. PSD95 and synaptophysin (Syn) are enriched in synaptosomes. **c** Subcellular redistribution of long-form caspase-2 in cultured mouse cortical neurons after NMDA treatment. Actin was used as internal loading control. Sample size: $n = 7$ for cytoplasm vs. nucleus comparison, $n = 3$ each for nuclear Casp-2L (Nuc-2L) after vehicle or NMDA treatment, $n = 3$ each for cytosolic Casp-2L (Cyt-2L) after vehicle or NMDA treatment, and $n = 3$ each for synaptosomal Casp-2L (Syn-2L) after vehicle or NMDA treatment. **d** NMDA-induced increase in caspase-2 dimerization in dendrites but not in the soma as revealed through pCasp2pro-VN and pCasp2pro-VC co-transfection. Veh, $n = 12$ neurons; NMDA, $n = 12$ neurons. **e** NMDA-induced increase in caspase-2 activity in dendrites but not in the soma as revealed through in situ measurement with VDVAD-FITC. Veh, $n = 14$ neurons; NMDA, $n = 13$ neurons. Data are expressed as mean ± SEM. Two-tailed Student's $t$ test: n.s. no significance, $*p < 0.05$, $**p < 0.01$, and $***p < 0.001$. Source data are provided as a Source Data file

reduced spine head diameters in cultured rat hippocampal neurons expressing control shRNA (Fig. 3a and Supplementary Fig. 3a) or treated with DMSO (Fig. 3b and Supplementary Fig. 3c). Interestingly, knocking down or inhibiting caspase-2 abolished NMDA-induced spine shrinkage in cultured hippocampal neurons (Fig. 3a, b and Supplementary Fig. 3b, d). These results suggest that caspase-2 is involved in either expression of LTD or LTD-induced spine shrinkage.

We further studied the role of caspase-2 in synaptic transmission in CA1 pyramidal neurons using brain slices from 3–4-week-old mice. Whole-cell voltage-clamp recordings of AMPAR-mediated miniature excitatory postsynaptic currents (mEPSCs), which reflect the response of the AMPAR to glutamate released spontaneously from a single synaptic vehicle, revealed that comparable amplitude and frequency of mEPSCs in WT and *Casp2* KO mice (Supplementary Fig. 3e). This observation indicates that caspase-2 deficiency does not affect the content of synaptic vehicles and probability of spontaneous glutamate release. We then examined evoked synaptic transmission by measuring paired pulse ratio (PPR) and input–output curves at the Schaffer collateral-CA1 synapses. PPR reflects the properties of presynaptic terminals from CA3 neurons, whereas input–output curves measure postsynaptic response to varying strengths of stimulation. Both PPR and input–output curves were indistinguishable between the two genotypes (Supplementary Fig. 3f, g), suggesting normal basal synaptic transmission.

*Casp2* KO mice displayed normal induction and expression of LTP at the Schaffer collateral-CA1 synapses (Fig. 3c). Interestingly, maintenance, but not induction, of LTD was impaired in

*Casp2* KO mice (Fig. 3d). This result indicates that LTD impairment is the reason why NMDA treatment does not induce spine shrinkage in cultured neurons when caspase-2 is knocked down or inhibited. Furthermore, we found that decay kinetics of synaptic transmission significantly differed between WT and *Casp2* KO mice. Faster decay kinetics were observed for both mEPSCs (Fig. 3e) and field excitatory postsynaptic potentials (fEPSPs; Fig. 3f) in *Casp2* KO hippocampal neurons, compared with WT neurons. Because mEPSCs are mediated by AMPARs, the change in decay time suggests that caspase-2 deficiency alters the composition of AMPARs.

**Caspase-2 is required for GluA1 internalization.** One major mechanism underlying LTD is internalization and subsequent degradation of synaptic AMPARs[49]. LTD impairment and abnormal EPSP decay kinetics in *Casp2* KO mice suggest that caspase-2 might regulate trafficking of AMPARs. We first examined if levels of AMPA and NMDA receptors were altered in *Casp2* KO mice. Compared with WT littermates, KO mice had higher levels of AMPAR subunit 1 (GluA1) in the hippocampus (WT: 100 ± 9% (mean ± SEM); KO: 141 ± 9%; $n = 5$ per group; $p < 0.05$ by two-tailed Student's $t$ test) without significantly altering levels of GluA2, GluA3, and NMDAR subunit 1 (GluN1) (Fig. 4a). The increase in GluA1 levels could result from either increased gene expression or reduced degradation. As we found that the hippocampal *Gria1* (encoding GluA1) mRNA level was comparable between the two genotypes (Fig. 4b), GluA1 degradation is impaired in *Casp2* KO mice.

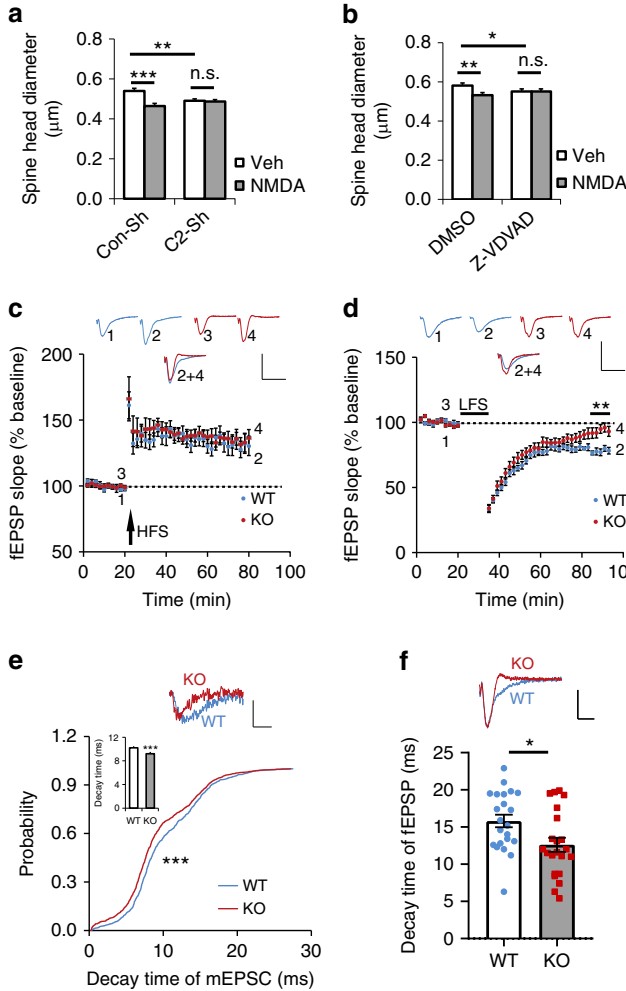

**Fig. 3** Caspase-2 deficiency impairs NMDA-induced spine shrinkage, diminishes LTD expression and alters decay kinetics. **a, b** Effects of caspase-2 knockdown or inhibition on spine head size in cultured rat hippocampal neurons. Neurons were transfected with constructs expressing Con-shRNA (Con-sh) or *Casp2* shRNA (C2-sh) on DIV14. Alternatively, the caspase-2 inhibitor Z-VDVAD-FMK (10 µM) or DMSO was added to neuronal cultures on DIV21. For visualization of spines, neurons were transfected with a construct expressing EGFP-actin on DIV14. On DIV28, neurons were treated with vehicle or 40 µM NMDA for 5 min and fixed 30 min after NMDA treatment for analysis of dendritic spines. Con-sh and Veh, 281 spines from 10 neurons; C2-sh and Veh, 292 spines from 10 neurons; Con-sh and NMDA, 178 spines from 6 neurons; C2-sh and NMDA, 356 spines from 12 neurons; DMSO and Veh, 369 spines from 13 neurons; Z-VDVAD and Veh, 308 spines from 11 neurons; DMSO and NMDA, 208 spines from 7 neurons; Z-VDVAD and NMDA, 299 spines from 10 neurons. **c** Normal LTP expression at Schaffer collateral-CA1 synapses of *Casp2* KO mice, as revealed by measurement of field excitatory postsynaptic potentials (fEPSP). LTP was induced with high-frequency stimulation (HFS). WT, 10 slices from 6 mice; KO, 11 slices from 7 mice. Scale bars, 1 mV (vertical) and 20 ms (horizontal). **d** Impaired LTD expression at Schaffer collateral-CA1 synapses of *Casp2* KO mice. LTD was induced with low-frequency stimulation (LFS). WT, 13 slices from 5 mice; KO, 14 slices from 8 mice. Scale bars, 1 mV (vertical) and 20 ms (horizontal). **e** Cumulative probability of mEPSC decay time in CA1 pyramidal neurons of WT and *Casp2* KO mice. Mean decay time was shown in the bar graph. WT, 840 events from 17 neurons in 5 mice; KO, 991 events from 20 neurons in 6 mice. Scale bars, 20 pA (vertical) and 10 ms (horizontal). **f** Decay time of fEPSP at CA1 pyramidal neurons. WT, 22 slices from 9 mice; KO, 22 slices from 11 mice. Scale bars, 0.5 mV (vertical) and 10 ms (horizontal). Data are expressed as mean ± SEM. Data were analyzed using two-tailed Student's *t* test, except **e** where Mann–Whitney test was used. n.s. no significance, *$p < 0.05$, **$p < 0.01$, and ***$p < 0.001$. Source data are provided as a Source Data file

A significant increase of GluA1 but no other AMPAR subunits could elevate the portion of homomeric GluA1 AMPARs. It has been reported that $Ca^{2+}$-permeable and GluA2-lacking AMPARs display a faster decay kinetics than heteromeric GluA1/GluA2 receptors[50,51]. Thus, it is possible that the observed faster decay kinetics of AMPARs in *Casp2* KO mice than WT mice (Fig. 3e, f) is due to an elevated level of homomeric GluA1 AMPARs. In support of this argument, we found that bath application of Naspm (50 µM), a selective blocker of GluA2-lacking AMPARs, erased the decay kinetics differences between WT and KO mice (Fig. 4c).

To examine GluA1 internalization and degradation directly, we stimulated cultured hippocampal neurons with 40 µM NMDA for 5 min to induce chemical LTD and internalization of surface AMPARs. As expected, NMDA treatment significantly reduced surface GluA1 and total GluA1 in cultured WT hippocampal neurons (Fig. 4d), indicating that synaptic GluA1 was internalized and degraded. However, in cultured *Casp2* KO hippocampal neurons, neither surface GluA1 nor total GluA1 was reduced after NMDA treatment (Fig. 4d). To confirm this finding, we used immunocytochemistry to visualize surface and internalized GluA1 in cultured hippocampal neurons. We found that NMDA treatment induced GluA1 internalization in WT neurons, but not in *Casp2* KO neurons (Supplementary Fig. 4). These results show that caspase-2 is essential for internalization of AMPARs upon LTD induction.

**Caspase-2 inhibits Akt signaling through cleavage of Rictor.** GSK3β is involved in internalization of AMPARs and required for

NMDAR-LTD[24,52]. In the hippocampus of *Casp2* KO mice, the inactive form of GSK3β (phosphorylated at Ser9, S9-GSK3β) was significantly increased, whereas total GSK3β (T-GSK3β) remained unchanged (Fig. 5a), suggesting that the activity of GSK3β was reduced. Consistently, Akt, the major upstream kinase of GSK3β[26], was more active in the *Casp2* KO hippocampus, as the level of the active S473-Akt (phosphorylated at Ser473) was significantly increased in *Casp2* KO mice compared with WT mice (Fig. 5a). These results indicate that an increase in Akt activity in the absence of caspase-2 attenuates the GSK3β activity, which leads to LTD impairment.

Because S473-Akt is a reliable readout for the activity of mTORC2[53,54], we sought to examine the impact of *Casp2* KO on the level of mTORC2. Rictor and mTOR are two important components of mTORC2, and their levels in the hippocampus were increased in *Casp2* KO mice (Fig. 5b), suggesting that caspase-2 normally inhibits the mTORC2 activity. Conversely, caspase-2 does not affect the activity of mTOR complex 1 (mTORC1), as the hippocampal level of phosphorylated S6K (pS6K), a downstream readout for mTORC1, was comparable between WT and KO mice (Supplementary Fig. 5a). In addition to mTORC2, PI3K also activates Akt through PDK1 and PDK2, which phosphorylate Akt at Thr308 and Ser473, respectively[55]. We found that the level of Thr308-Akt was not elevated in the hippocampus of *Casp2* KO mice (Supplementary Fig. 5a). Furthermore, the abundance of the catalytic subunit of PI3K (p110), PDK1, and PDK2 was normal in *Casp2* KO mice (Supplementary Fig. 5a). These results indicate that caspase-2 specifically regulates the Akt activity through mTORC2.

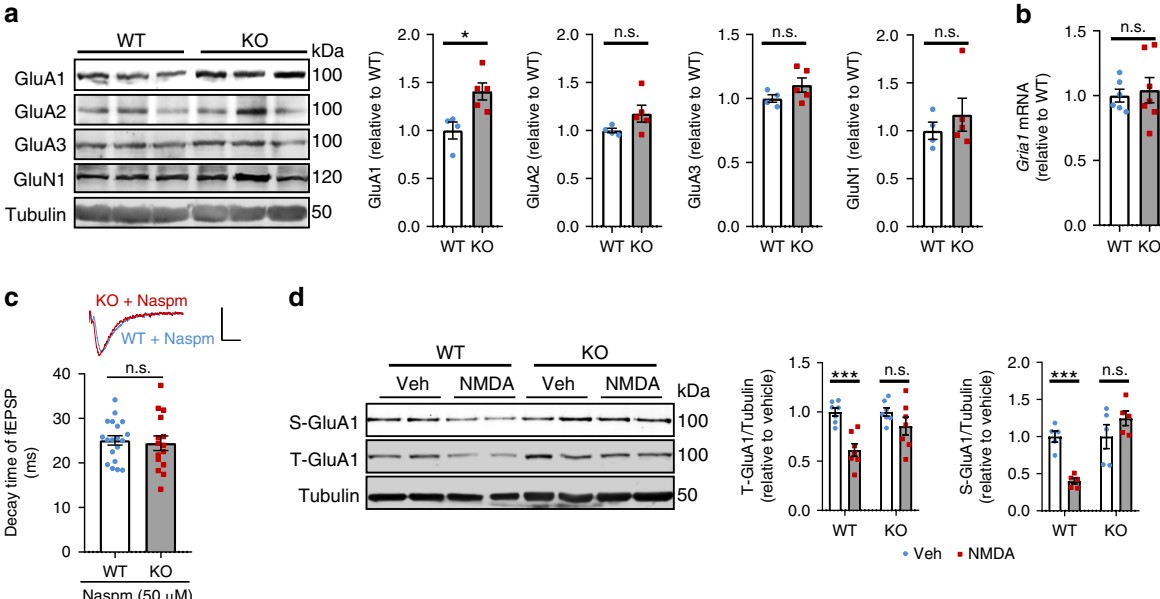

**Fig. 4** Caspase-2 is required for GluA1 internalization and degradation. **a** Immunoblotting analysis and quantification of glutamate receptors in WT and *Casp2* KO hippocampi. WT, $n = 4$; KO, $n = 5$. **b** Levels of *Gria1* mRNA in WT and *Casp2* KO hippocampi as revealed by quantitative RT-PCR with 18 s rRNA as internal control. WT, $n = 6$; KO, $n = 7$. **c** Decay time of fEPSP at CA1 pyramidal neurons in the presence of Naspm. WT, 19 slices from 3 mice; KO, 15 slices from 3 mice. Scale bars, 0.5 mV (vertical) and 10 ms (horizontal). **d** Immunoblotting analysis and quantification of surface and total GluA1 in cultured mouse hippocampal neurons treated with vehicle or NMDA. $n = 5$–7. Data are expressed as mean ± SEM. Two-tailed Student's *t* test: n.s. no significance, $*p < 0.05$ and $***p < 0.001$. Source data are provided as a Source Data file

To further investigate the regulation of mTORC2 by caspase-2, we used lentivirus to overexpress caspase-2L in hippocampal neurons and then analyzed the mTORC2-Akt-GSK3β signaling cascade. Immunoblotting analysis showed that the amount of full-length Rictor (200 kDa) was dramatically reduced (Fig. 5c), whereas two new Rictor protein bands at ~90 and 110 kDa appeared in lysates of caspase-2L-overexpressing neurons (Fig. 5c). The level of endogenous Rictor is low in neurons (Fig. 5c), and levels of some Rictor breakdown products might be too low to be detected in neuronal extracts. To further explore caspase-2 mediated Rictor degradation, we co-transfected plasmids containing Rictor-Myc and caspase-2L sequences into HEK293 cells. Co-expression of caspase-2 and Rictor produced several Rictor polypeptides (Supplementary Fig. 5b), as detected by Rictor antibodies (recognizing the N-terminal region of Rictor) and Myc antibodies (recognizing the Myc tag at the C terminus of Rictor). Taken together, these results suggest that caspase-2 degrades Rictor at multiple cleavage sites. In agreement with this conclusion, caspase-2L overexpression reduced the activity of Akt and the level of S9-GSK3β (Fig. 5d), likely due to disassembly of mTORC2 in the event of Rictor degradation. Unexpectedly, caspase-2L overexpression greatly increased the level of pS6K, although it significantly reduced the level of mTOR (Fig. 5d). This could result from mTORC1 upregulation due to increased availability of components shared by the two mTOR complexes when caspase-2 disrupts mTORC2 formation.

The above results suggest that caspase-2 could reduce the abundance of mTORC2 by degrading Rictor, which will increase the GSK3β activity by reducing Akt activation and thereby promote LTD. We tested this hypothesis in *Casp2* KO neurons in which chemical LTD-inducing NMDA treatment fails to induce internalization of AMPARs (Fig. 4d and Supplementary Fig. 4). NMDA treatment reduced levels of Rictor, active S473-Akt and inactivated S9-GSK3β in WT neurons, but not *Casp2* KO neurons (Fig. 5e). Interestingly, we noticed that levels of Rictor, S473-Akt and S9-GSK3β were comparable in cultured WT and *Casp2* KO

neurons (Supplementary Fig. 5c), whereas they were increased in *Casp2* KO hippocampal tissues (Fig. 5a, b). The discrepancy could result from differences in the regulation of Rictor-Akt-GSK3β signaling by caspase-2 in vivo and in vitro. The regulation is activity-dependent, but cultured primary neurons are generally "quiet". Similarly, we observed no differences in total GluA1 (Supplementary Fig. 5c) or surface GluA1 (Supplementary Fig. 4) between WT and *Casp2* KO neuronal cultures. Collectively, we showed that caspase-2 negatively regulates Rictor-Akt-GSK3β signaling in an activity-dependent manner to alter levels of AMPARs.

Based on the aforementioned results, LTD induction could lead to a series of sequential events, including caspase-2 activation, Rictor degradation, Akt inhibition, GSK3β disinhibition, internalization of AMPA receptors and spine elimination. This signaling cascade would predict that altering Rictor levels will change density of dendritic spines. Indeed, we found that reducing Rictor levels with shRNA (Fig. 5f) or caspase-2 overexpression (Supplementary Fig. 5d) significantly decreased spine density. On the contrary, Rictor overexpression increased spine density, and this increment was abolished by caspase-2L co-expression (Supplementary Fig. 5d).

**Caspase-2 deficiency elevates anxiety and enhances memory.** Deficiency of caspase-2 was reported to improve memory deficits in mouse models for Alzheimer's disease[37,38] and Huntington's disease[39]. However, it is still unknown whether caspase-2 modulates behaviors in normal mice. We performed a battery of behavioral tests to examine anxiety and memory in *Casp2* KO mice and their WT littermates. WT and KO mice traveled comparable distances at similar velocities in open field tests (Supplementary Fig. 6a, b). Furthermore, KO mice performed normally on rotarod tests (Supplementary Fig. 6c). These results indicate that KO mice have normal motor function and motor learning. However, in the first 5 min of the open field test when

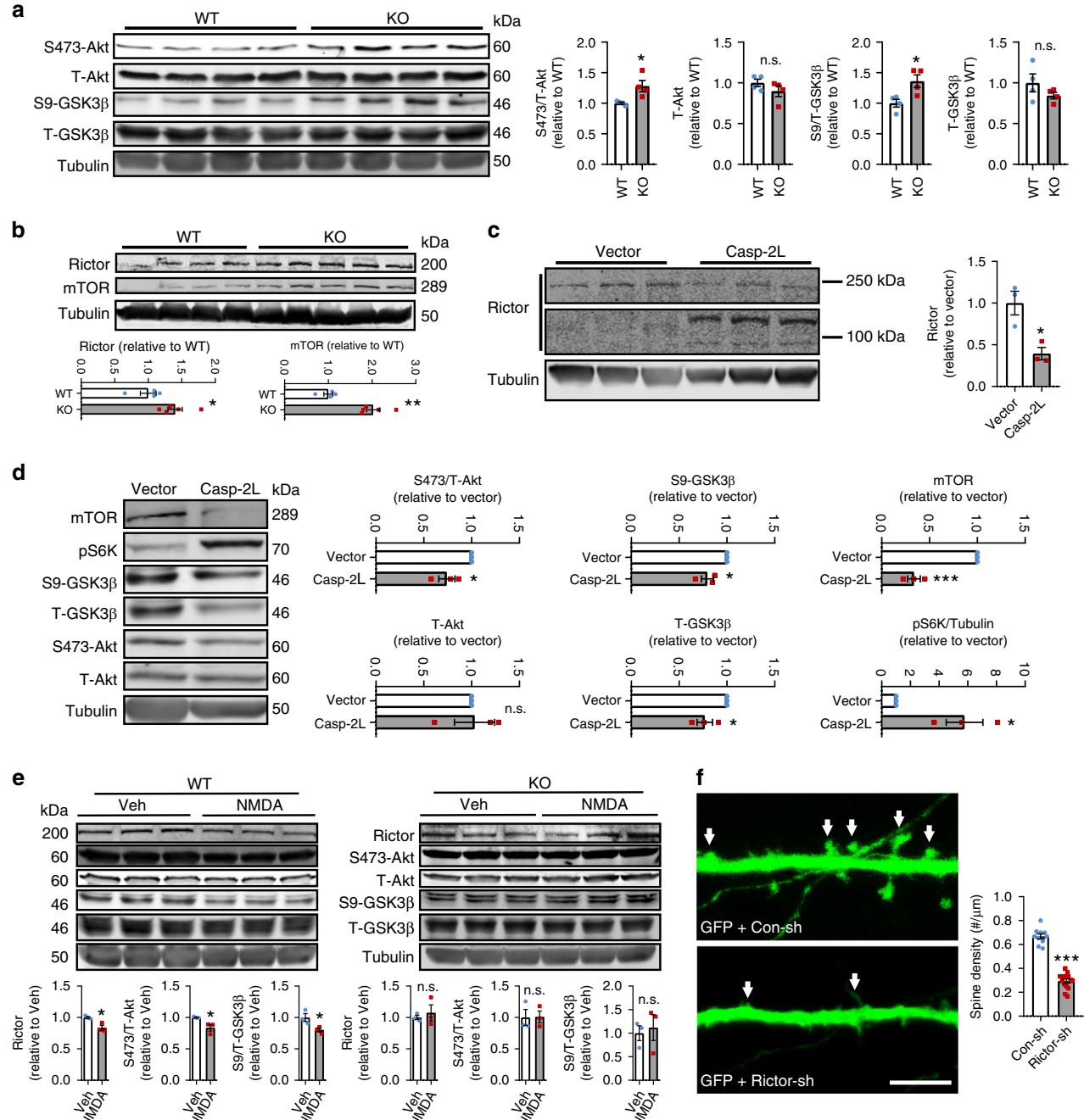

**Fig. 5** Caspase-2 regulates the Akt-GSK3β signaling cascade through cleavage of Rictor. **a** Immunoblotting analysis and quantification of total Akt, Ser473-Akt, total GSK3β, and Ser9-GSK3β in the hippocampus of WT and *Casp2* KO mice. Tubulin was used as a loading control. $n = 4$ per genotype. **b** Levels of Rictor and mTOR in the hippocampus of WT and *Casp2* KO mice. WT, $n = 4$; KO, $n = 5$. **c** Cleavage of Rictor in cultured mouse hippocampal neurons infected with lentivirus expressing caspase-2L. Note that full-length Rictor is 200 kDa, whereas the two cleavage products are above and below 100 kDa, respectively. **d** Immunoblotting analysis and quantification of total Akt, Ser473-Akt, total GSK3β, Ser9-GSK3β, mTOR, and pS6K in cultured mouse hippocampal neurons. Neurons on DIV6 were infected with lentivirus expressed GFP or caspase-2L and lysed 96 h later. $n = 3$ for each viral vector. **e** Immunoblotting analysis and quantification of Rictor, total Akt, Ser473-Akt, total GSK3β, and Ser9-GSK3β in cultured WT and *Casp2* KO mouse hippocampal neurons treated with vehicle or NMDA. $n = 3$. **f** Knockdown of Rictor with shRNA (Rictor-sh) reduced spine density in cultured hippocampal neurons compared with control shRNA (Con-sh). Con-sh, $n = 10$ neurons; Rictor-sh, $n = 14$ neurons; Scale bar, 5 µm. Data are presented as mean ± SEM. Two-tailed Student's *t* test: n.s. for $p > 0.05$, $*p < 0.05$, $**p < 0.01$, and $***p < 0.001$. Source data are provided as a Source Data file

mice were adapting to the testing environment, KO mice spent less time and traveled less in the center zone than WT mice (Fig. 6a, b), indicating that KO mice had an elevated level of anxiety. This anxiety-like behavior was confirmed in light–dark box tests, as KO mice entered the light chamber fewer times compared with WT mice (Fig. 6c).

We started to examine a potential role of caspase-2 in memory using T maze alteration paradigm. Both WT and KO mice performed similarly in the test (Supplementary Fig. 6d), indicating that *Casp2* KO mice have normal working memory. Strikingly, in contextual fear conditioning tests KO mice spent more time in freezing 24 h after training (Fig. 6d), indicating the

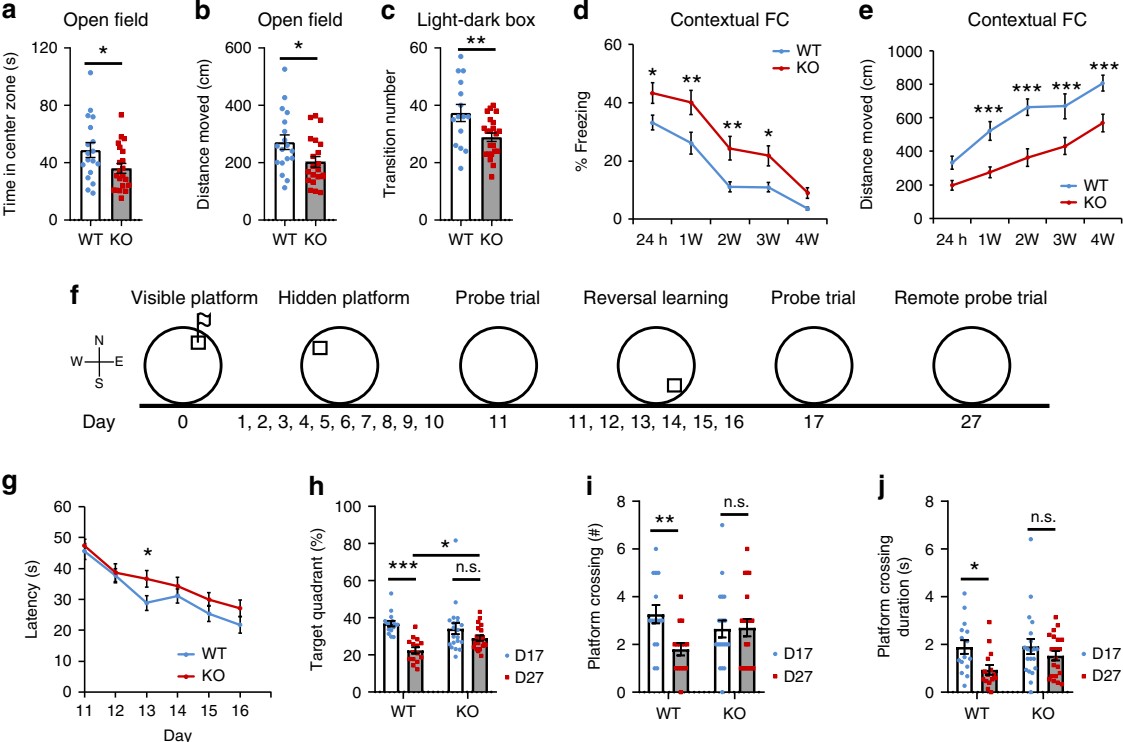

**Fig. 6** *Casp2* KO mice display elevated anxiety, enhanced fear memory and impaired cognitive flexibility. **a, b** Time spent and distance traveled in the center zone in open field tests. $n = 18$ WT mice and 21 KO mice. **c** Transition numbers between two chambers in light–dark exploration tests. $n = 15$ WT mice and 22 KO mice. **d, e** Percentage of time spent in freezing and distance traveled in contextual fear conditioning (FC) tests. $n = 18$ WT mice and 22 KO mice. **f** Schedule for training sessions and probe trials in water maze tests. **g** Latency for mice to locate the hidden platform during reversal spatial learning. $n = 15$ WT mice and 21 KO mice. **h** Percentage of time spent in the target quadrant during the 2nd probe trial and the remote probe trial. $n = 15$ WT mice and 20 KO mice. **i** Number of time when mice cross the location of the hidden platform during the 2nd probe trial and the remote probe trial. $n = 15$ WT mice and 20 KO mice. **j** Performance at the probe trials. Duration spent in crossing the location of the hidden platform was significantly shorter on day 27 than on day 17 in WT mice but not KO mice. Two-tailed Student's *t* test for **a**–**c**. Two-way ANOVA with Bonferroni post hoc test for **d**–**j**. *$p < 0.05$, **$p < 0.01$ and ***$p < 0.001$. Data are presented as mean ± SEM. Source data are provided as a Source Data file

long-term memory was enhanced. Notably, KO mice maintained elevated fear memory for at least 3 weeks after the initial conditioning training (Fig. 6d). In agreement with increased freezing behavior, KO mice also traveled shorter distances than WT mice in the fear conditioning chamber during memory retrieval tests (Fig. 6e). In contrast, WT and KO mice had indistinguishable cued fear conditioning memory (Supplementary Fig. 6e).

We used the Morris water maze (MWM) to test spatial and reference memory in mice. *Casp2* KO mice performed normally in the visible platform of the MWM test (Supplementary Fig. 6f, g), indicating that vision and swimming ability are not affected in KO mice. In the hidden platform version of the MWM test, WT and KO mice had similar swimming path length and escape latency during training (Supplementary Fig. 6h, i). On day 11 probe trial, WT and KO mice spent comparable amounts of time in the target quadrant that was used to harbor the hidden platform (Supplementary Fig. 6j). They also crossed the platform location with similar numbers and duration (Supplementary Fig. 6k, l). These results show that caspase-2 deficiency does not alter spatial memory. Because LTD expression was impaired in *Casp2* KO mice (Fig. 3d) and hippocampal LTD has been proposed to mediate reversal learning in the MWM test[56], we continued the spatial memory test with reversal training by moving the hidden platform to the opposite quadrant (Fig. 6f). Indeed, *Casp2* KO mice showed learning deficits in the reversal hidden platform training and displayed prolonged escape latency in the third reversal training sessions (Fig. 6g). Although WT and

KO mice had comparable memory retrieval ability in the probe trial conducted 1 day after the last reversal training as they spent similar amounts of time in the target quadrant and the location of the hidden platform (Supplementary Fig. 6m–o), a remote probe trial showed that spatial memory decayed at a slower rate in KO mice than in WT mice. The remote probe trial was carried out 10 days after the 2nd probe trial (Fig. 6f). Time spent in the target quadrant (Fig. 6h), number of crossings from the platform location (Fig. 6i) and duration spent in the platform location (Fig. 6j) in the remote probe trial were significantly lower than those in the 2nd probe trial for WT mice, but not for KO mice. These behavioral results indicate that *Casp2* KO mice have cognitive inflexibility, but more enduring spatial memory compared with WT mice.

## Discussion
Our study reveals a signaling pathway caspase-2–mTORC2–Akt–GSK3β, where caspase-2 inhibits mTORC2 via cleavage of Rictor, leading to a reduction in the Akt activity and eventually an increase in the GSK3β activity. As previous studies show a requirement of the GSK3β activity for AMPAR internalization and LTD induction[24,52] and this study demonstrates deficits in NMDA-induced AMPAR internalization and LTD in the absence of caspase-2, this signaling pathway should be essential for removal of synaptic AMPARs and expression of NMDAR-LTD. This argument could be strengthened by mutagenesis studies validating that Rictor is a substrate for caspase-2. We also observed that caspase-2 deficiency results in spine pruning

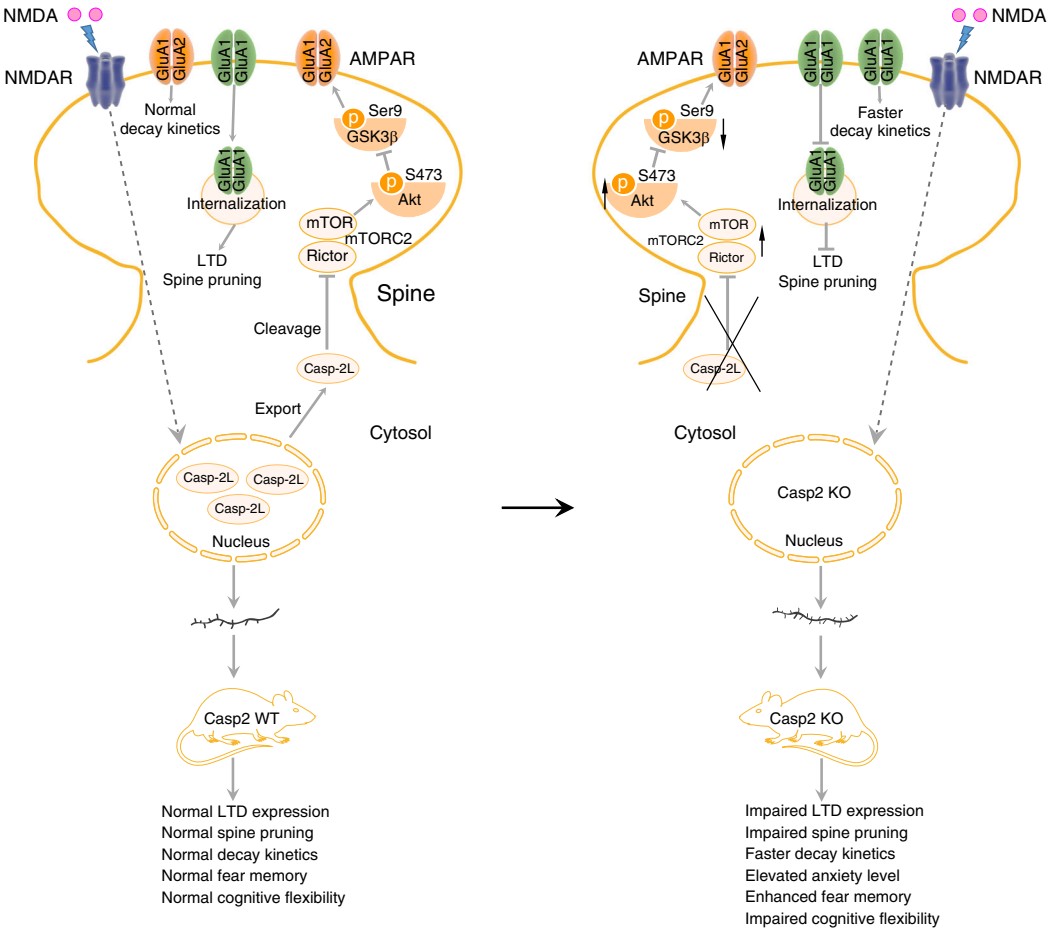

**Fig. 7** A model of caspase-2 mediated synaptic plasticity and cognitive flexibility. Under basal condition, the majority of caspase-2L is localized in the nucleus. Upon NMDA stimulation, caspase-2L is exported from the nucleus into dendrites, where it mediates the cleavage of mTORC2 scaffold protein Rictor. Decreased mTORC2 level reduces the Akt activity but increases the GSK3β activity, therefore inducing AMPAR internalization and LTD. Caspase-2 deficiency impairs LTD and spine pruning, increases GluA1 level, causes faster decay kinetics, elevates anxiety level, and impairs cognitive flexibility

impairment and cognitive inflexibility. Given that deficits in LTD have been associated with synapse pruning[15,30,31] and cognitive inflexibility[56,57], we propose that caspase-2 diminishes synaptic strength by disrupting mTORC2, which is essential for synapse elimination and cognitive flexibility (Fig. 7).

Caspases are divided into initiator caspases (e.g., caspase-2 and caspase-9) and executioner caspases (e.g., caspase-3). Caspase 3 has been shown to be essential for AMPAR internalization, LTD and spine pruning[30–32]. Our study indicates that caspase-2 is also required for this spine pruning process. These findings establish a role for caspases in elimination of excess synapses during the development of the nervous system in addition to their well-known role in apoptosis. Despite of similar roles of these two caspases in spine pruning and AMPAR internalization, they employ distinct signaling pathways to execute them. We found that caspase-2 activation leads to Rictor degradation, which should cause disruption of the mTORC2 complex and degradation of its components. Indeed, our results show that caspase-2 overexpression reduces neuronal mTOR levels, whereas caspase-2 ablation does the opposite. mTORC2 disruption will reduce Akt activation[26], leading to GSK3β activation and AMPAR internalization[24]. Conversely, caspase-3 directly reduces the Akt activity by cleaving Akt[30]. Although caspase-2 could facilitate caspase-3 activation to induce apoptosis in some circumstances[58], it is unlikely that brief NMDA treatment inhibits Akt by activating caspase-3 via caspase-2 because we found that

caspase-2 deficiency or overexpression had no effect on levels of total Akt.

We found that reducing Rictor levels with shRNA or caspase-2 overexpression significantly decreased spine density in cultured neurons. This finding is consistent with a previous report that abolishment of Rictor expression reduces spine density in hippocampal CA1 pyramidal neurons and impairs long-term memory consolidation[27]. Furthermore, we found that Rictor overexpression increased spine density in cultured hippocampal CA1 neurons. These results indicate that mTORC2 bidirectionally regulates structural plasticity of synapses, which could be an underlying mechanism for the role of mTORC2 in learning and memory. Interestingly, one study reported that mTORC2 is also required for hippocampal mGluR-LTD[28]. This finding appears to contradict our observation that caspase-2 promotes hippocampal NMDAR-LTD by disrupting mTORC2. The opposing action of mTORC2 on these two forms of LTD may result from distinct molecular mechanisms that underlie NMDAR-LTD and mGluR-LTD. For example, hippocampal NMDAR-LTD, but not mGluR-LTD, requires protein phosphatases[59], which may target Ser9-GSK3β. Given that little is known about upstream regulators mTORC2, our discovery that caspase-2 cleaves Rictor should help elucidate the role of mTORC2 in physiological processes in addition to synaptic plasticity. It has been reported that lack of nutrients causes metabolic decline and caspase-2 activation[60]; however, it is unclear how caspase-2 senses nutrients to modulate cellular metabolism. One possibility

is that caspase-2 alters the metabolic homeostasis by regulating the activity of mTORC2, which serves as a major effector of the insulin/IGF-1 signaling pathway[61].

We found that the majority of caspase-2 is localized in the neuronal nucleus. Consistent with findings in HeLa cells[62], the BiFC assay indicates that caspase-2 is activated in the cytoplasm but not the nucleus of a neuron. While the BiFC signal is only induced in HeLa cells in response to certain stresses[62], we detected the BiFC signal in the cytoplasm in vehicle-treated neurons. Additional studies are needed to investigate the possibility that neurons may contain active caspase-2 in some specific cytoplasmic compartments under the basal condition. We found that a brief NMDA treatment led to translocation of some caspase-2 from the nucleus to the cytoplasm. It has been shown that signals from synapses can be conveyed to the nucleus through electrochemical signaling, calcium waves, and physical translocation of signaling proteins[63]. Once the nucleus receives the synaptic signals, it promotes activity-dependent gene expression over a time course of hours. This communication between synapses and the nucleus is essential for synaptic plasticity and neural development. Our results show that caspase-2 is physically translocated from the nucleus to the cytoplasm within minutes after NMDAR activation, revealing a fast mode of nuclear response to synaptic signals independent of gene expression. This translocation is not secondary to apoptosis, because we found that the number of cells positive for cleaved caspase-3 was comparable between vehicle- and NMDA-treated neuronal cultures. Because procaspase-2 is partially active upon dimerization[44], this nucleus-to-cytoplasm translocation could be the main reason why the activity of caspase-2 is increased in dendrites after activation of NMDA receptors. It would be interesting to investigate whether caspase-2 disrupts mTORC2 at or outside synapses to promote AMPAR internalization in future studies.

GluA1/A2 heteromers are the dominant AMPARs at CA1 neuronal synapses[64], although some studies found the presence of a small population of homomeric GluA1 AMPARs in the hippocampus[65,66]. In Casp2 KO mice, we detected a significant increase of GluA1 but no other AMPAR subunits, suggesting that the portion of homomeric GluA1 AMPARs might be increased. In support of this inference, we detected faster decay kinetics in Casp2 KO mice. It has been reported that $Ca^{2+}$-permeable and GluA2-lacking AMPARs display a faster decay kinetics than heteromeric GluA1/GluA2 receptors[50,51]. Furthermore, we found that Naspm, a blocker of GluA2-lacking AMPARs, abolished the difference in decay kinetics between WT and Casp2 KO mice. This interesting observation suggests that caspase-2 may serve as an indispensable regulator to control the temporal fidelity of synaptic transmission. Surprisingly, though spine density increases significantly, the frequency and amplitude of mEPSCs remain unchanged in Casp2 KO mice. This could be due to the fact that spine density is increased in distal, but not proximal, dendrites of Casp2 KO neurons. The mEPSCs recorded in the soma are not sensitive enough to detect local dendritic depolarization produced by distal synapses[67,68]. A previous study reported no difference in spine density in hippocampal CA1 pyramidal neurons between WT and Casp2 KO mice[37], and the study might focus on dendritic spines in proximal dendrites.

In agreement with previous studies[37,38], we found that Casp2 KO mice had normal learning ability in the acquisition phase of the MWM test. These mice, however, displayed prolonged latency to locate the new platform position in the reversal acquisition phase, indicating impaired cognitive flexibility. Interestingly, Casp2 KO mice showed slower memory loss in the remote memory retrieval test than WT littermates, which may be related to the LTD deficit, a physiological process that has been suggested

to underlie forgetting[69,70]. Inability to erase remote memory traces should contribute to the impairment in cognitive flexibility. Taken together, our study reveals a critical physiological role for caspase-2 in cognitive flexibility and the underlying molecular and synaptic mechanism.

## Methods

**Animals.** Casp2 KO mouse strain (stock no: 007899) was obtained from the Jackson Laboratory and described previously[40]. All mice were maintained on a 12-/12-h light/dark cycle with ad libitum access to water and food. Pregnant Sprague Dawley rats were purchased from Charles River Laboratories. All animal procedures were approved by the Scripps Florida Institutional Animal Care and Use Committee (protocol # 16-003).

**Primary neuronal culture and transfection.** Hippocampal neurons were cultured from E18.5 Sprague Dawley rat embryos or P0 newborn mice according to procedures described previously[17]. Briefly, isolated hippocampi were removed and digested with 20 U/ml of papain in Hank's Balanced Salt Solution at 37 °C for 30 min. Dissociated neurons were grown in Neurobasal media (Invitrogen) supplemented with 2% B27, 1% GlutaMAX (Gibco) and 1% penicillin–streptomycin at 37 °C and 5% $CO_2$ incubator. For transfection, 2 μl of Lipofectamine 2000 (Invitrogen) and plasmid DNA (0.4 μg/kb) were added to 100 μl of Neurobasal medium, respectively, and incubated at room temperature for 5 min. The two parts were then mixed together and incubated at room temperature for an additional 25 min before adding to the neurons.

**Z-VDVAD-FMK treatment of cultural neurons.** Cultured rat hippocampal neurons were transfected with a construct expressing enhanced green fluorescent protein (EGFP)-actin on DIV14. Neurons on DIV21 were then treated with Z-VDVAD-FMK (20 μM, R&D Systems #FMK003) or DMSO for 7 days. Neurons were fixed and stained with an anti-EGFP antibody on DIV28.

**Knockdown of caspase-2 and Rictor with shRNA.** To knock down caspase-2, we targeted the following sequence as described previously[71]: GCCGAGAATGTGG AACTCCT. For Rictor knockdown, we targeted the sequence GCCAGTAAGAT GGGAATCATT as described previously[72]. The DNA oligonucleotides containing the shRNA target sequence, a 9-nucleotide loop region (TTCAAGAGA), and the shRNA antisense sequence were cloned into tdTomato-expressing plasmid pll3.7[73]. A scrambled shRNA sequence was used as a control.

**Caspase-2 knockdown rescue.** To generate constructs expressing shRNA-resistant caspase-2L and caspase-2S (in which mRNA includes exon 9 sequence and incorporates an in-frame stop codon, producing a truncated protein containing the first 291 amino acids of the caspase-2L protein), two silent mutations were introduced into the region targeted by the Casp2 shRNA (GCCAGAATG T**GGAA**CTCCT was mutated to GCCAGAATGT**T**GA**G**CTCCT, where the mutated sites were highlighted in bold). On DIV14, rat hippocampal neurons were co-transfected with constructs expressing EGFP-actin, Casp2 shRNA and shRNA-resistant long or short caspase-2 (R2L or R2S). Nearly all transfected neurons contained the three constructs. On DIV28, neurons were fixed and stained with an antibody against EGFP.

**Rictor-Myc-expressing construct.** The mouse Rictor coding sequence extended at its 3′ end with a Myc-coding sequence (gccGAACAAAAACTCATCTCAGAAGA GGATCTGaatagctag, where the Myc-encoding sequence is listed in capital letters) was cloned into pUltra (a gift from Malcolm Moore; Addgene plasmid #24129).

**NMDA treatment.** Unless specified, rat or mouse hippocampal neurons were treated with 40 μM NMDA for 5 min and then washed with culture medium for three times. After cultural medium was added, neurons were returned to incubator for another 30 min before harvest.

**Production of lentivirus.** The coding sequence of caspase-2L was cloned into pUltra (a gift from Malcolm Moore; Addgene plasmid #24129). Lentivirus was produced using a standard calcium phosphate transfection protocol. In brief, HEK293 cells were plated in 10-cm dishes coated with poly-D-lysine. Plasmids expressing caspase-2L (or GFP only as a control), pMDL (packaging plasmid expressing gag and pol), pVSVG (packaging plasmid expressing envelope protein VSVG), and pREV (packaging plasmid expressing Rev) were co-transfected into cells. After 18–21 h of transfection, medium was replaced with fresh DMEM supplemented with 2% FBS and 1% glutamate. Media containing lentivirus were collected in the next 2 days and stored at −80 °C.

**Immunocytochemistry.** Cultured neurons were fixed for 20 min with 4% paraformaldehyde and 4% sucrose at room temperature. Cells were washed with phosphate-buffered saline (PBS) and permeabilized with 0.25% Triton X-100 in

PBS. After being washed three times with PBS, neurons were incubated with blocking buffer (PBS containing 5% BSA and 0.1% Triton X-100) for 1 h at room temperature. Afterwards, cells were incubated with the following primary antibodies in blocking buffer at 4 °C overnight: rabbit anti-GFP (Clontech #632592, 1:4000), rabbit anti-cleaved caspase-3 (Cell Signaling Technology #9661, 1:1000) and mouse anti-HA (Covance #MMA-101P, 1:500). Appropriate DyLight conjugated secondary antibodies (Jackson ImmunoResearch Laboratories) were used after primary antibodies were washed off with PBS for three times. Nuclei were counterstained with DAPI (Invitrogen #D1306, 10 mg/ml stock solution, 1:10000 dilution). Images were acquired using a Nikon C2+ confocal microscope.

**BiFC assay.** Plasmids pCasp2pro-VN and pCasp2pro-VC were kindly provided by Drs Sally Kornbluth and Kenkyo Matsuura[45]. Mouse hippocampal neurons were transfected with pCasp2pro-VN and pCasp2pro-VC on DIV7. Forty-eight hours post transfection, neurons were treated with NMDA (40 μM) or vehicle for 5 min, washed with culture medium twice, and fixed for MAP2 staining (rabbit anti-MAP2, Santa Cruz Biotechnology #sc-20172, 1:200). We noticed some transfected cells with high levels of Venus fluorescence but without MAP2 immunoreactivity, and these cells were not included in our analysis.

**RNA isolation and qRT-PCR.** Hippocampal total RNA was isolated from adult WT and Casp2 KO mice using TRIzol Reagent (Invitrogen) according to the manufacturer's instruction. To remove genomic DNA contamination, total RNA samples were treated with RNase-free DNase for 30 min at 37 °C, followed by phenol/chloroform extraction. cDNA was synthesized using M-MuLV Reverse Transcriptase (NEB #M0253). Real-time PCR was performed with specific gene primers using the Power SYBR Green PCR master mix (Roche #04913914001), and gene expression was presented relative to 18S rRNA using the $2-\Delta\Delta Ct$ method. Sequences of PCR primers are as follows: 5-ctcgcccttgtcgtaccac-3 (forward) and 5-gtccgccctgagaaatccag-3 (reverse) for Gria1; 5-ccgcagctggaataatgga-3 (forward) and 5-ccctcttaatcatggcctca-3 (reverse) for mouse 18S rRNA.

**Western blotting.** Hippocampal tissues or cultured primary neurons were lysed on ice for 30 min in lysis buffer containing 10 mM Tris (pH 7.4), 1% Triton X-100, 150 mM NaCl, 10% glycerol, and freshly added protease inhibitors (Roche Complete Protease Mini, #4693159001) and phosphatase inhibitors (PhosStop pellets, Sigma Aldrich, #4906845001). Lysates were centrifuged at $15,000 \times g$ for 30 min at 4 °C, and supernatants were saved as protein extracts. Protein concentration in extracts was determined using Lowry Protein Assay (Bio-Rad). Protein samples were run on SDS-PAGE gels and transferred to PVDF membrane. Membrane was blocked with Odyssey Blocking Buffer (Fisher Scientific). The following primary antibodies were used: mouse anti-α-tubulin (Sigma-Aldrich #T6074, 1:10000), mouse anti-PSD95 (Thermo Fisher Scientific #MA1-045; 1:1000), rabbit anti-caspase-2 (Abcam #ab179519, 1:2000), mouse anti-β-actin (Sigma-Aldrich #A5441, 1:10000), mouse anti-synaptophysin (Abcam #ab8049, 1:1000), rabbit anti-GluA1 (Millipore #AB1504, 1:1000), mouse anti-GluA2 (Millipore #MABN71, 1:1000), mouse anti-GluA3 (Millipore #MAB5416, 1:1000), mouse anti-GluN1 (Millipore #MAB363, 1:1000), mouse anti-PDK1 (Santa Cruz Biotechnology #sc-293160, 1:1000), mouse anti-PDK2 (Santa Cruz Biotechnology #sc-517284, 1:1000), mouse anti-p110 (Santa Cruz Biotechnology #sc-8010, 1:500), mouse anti-Rictor (Santa Cruz Biotechnology #sc271081, 1:500), rabbit anti-S473-Akt (Cell Signaling Technology #2920, 1:1000), rabbit anti-T308-Akt (Cell Signaling Technology #9275, 1:1000), mouse anti-pan Akt (Cell Signaling Technology #9271, 1:1000), rabbit anti-GSK3β (Cell Signaling Technology #9315, 1:1000), rabbit anti-Ser9-GSK3β (Cell Signaling Technology #9331, 1:1000), rabbit anti-mTOR (Cell Signaling Technology #2972, 1:1000), rabbit anti-Histone H3 (Cell Signaling Technology #4499, 1:1000), and rabbit anti-pS6K (Cell Signaling Technology #9205, 1:1000). Appropriate IRDye infrared secondary antibodies (LI-COR Biosciences) were used at a dilution of 1:10000. Odyssey Infrared Imaging System (LI-COR Biosciences) was used to detect the signal of target proteins. Uncropped western blots are included in the Source Data file.

**Time-lapse imaging of dendritic spines.** Rat hippocampal neurons were grown on glass-bottomed 35 × 35-mm dishes. On DIV14, neurons were co-transfected with a construct expressing EGFP-actin and a construct expressing either control shRNA or Casp2 shRNA. Between DIV21-28, time-lapse imaging was performed for 2 h using a 60× oil immersion objective under a Nikon C2+ confocal microscope (Nikon Instruments Inc.) equipped with a stage-top chamber (INUG2A-TIZ, Tokai Hit Co.). The chamber was humidified and maintained at 37 °C with 5% $CO_2$.

**Surface GluA1 staining and internalization assay.** Mouse hippocampal neurons on DIV12 were incubated with an antibody against GluA1 (Millipore #PC246, 1:20) for 10 min at 37 °C and then stimulated with 40 μM NMDA or vehicle for 5 min. Neurons were fixed with 4% formaldehyde immediately after the stimulation, and surface-remaining GluA1 was saturated by incubation with an Alexa Fluor 488-conjugated secondary antibody. Neurons were permeabilized, and internalized GluA1 was stained with an Alexa Fluor 594-conjugated secondary antibody.

**Golgi staining.** Two-month-old WT and Casp2 KO mice were selected for Golgi staining using the FD Rapid GolgiStain Kit (FD Neurotechnologies, Inc.) as described previously[17]. We used Neurolucida software (Microbrightfield Inc) to trace primary dendrites of Golgi-impregnated CA1 pyramidal neurons under a Nikon Eclipse E800 microscope equipped with a motorized stage. The position of each dendritic spine was marked along a dendrite. We traced 3–6 CA1 neurons in a mouse. The average of each measurement from these neurons was used as the value of the mouse. Dendritic length and spine density of each traced neuron were calculated using NeuroExplorer software (MicroBrightField Inc). The experimenter was blind to the genotype.

**Surface biotinylation assay.** Surface biotinylation experiments were performed as previously described[19]. Briefly, mouse hippocampal neurons on DIV12 were washed with PBS twice and incubated with 0.25 mg/ml of Sulfo-NHS-SS-Biotin (Pierce) for 15 min on ice. Then, neurons were rinsed twice with ice-cold 50 mM Tris-Cl, pH 7.4 to remove free biotin. Neurons were lysed in radioimmune precipitation assay buffer for 30 min followed by centrifugation at 13,000 rpm for 30 min. One-third of the supernatant was saved to determine the total level of GluA1. To isolate biotin-labeled (surface) GluA1, the other two-thirds of the supernatant were incubated with Streptavidin Sepharose beads (GE Healthcare) overnight at 4 °C. Resin was then washed three times and eluted with 2× SDS sample buffer at 96 °C for 15 min followed by western blot analyses.

**Synaptosome preparation.** Brains from 4-week-old mice were homogenized in 10 volumes of buffer containing 1 mM EDTA, 5 mM Tris-Cl (pH 7.4), 0.32 M sucrose, and protease/phosphatase inhibitor cocktail (Roche). Homogenates were subjected to centrifugation at 1000 $g$ for 10 min. The supernatant was collected and layered on the top of 1.2 M sucrose and centrifuged at 160,000 $g$. The interface was collected and layered on the top of 0.8 M sucrose and centrifuged again. Synaptosomes were pelleted at the bottom and resuspended for immunoblotting analysis.

**Electrophysiology.** Mice at P21-P28 were used for electrophysiological recording. Mouse was decapitated under isoflurane anesthesia. Brain was rapidly removed and placed in ice-cold artificial cerebrospinal fluid (aCSF) containing (in mM) 124 NaCl, 3 KCl, 26 NaHCO₃, 1.25 NaH₂PO₄, 1 MgSO₄, 2 CaCl₂, and 10 D-glucose, equilibrated with 95% O₂ and 5% CO₂. Hippocampal coronal slices (350 μm) were obtained using a vibratome (Leica VT 1200 s, Germany) and then transferred to oxygenated aCSF at 32 °C for recovery.

*Whole-cell patch-clamp recordings:* Slices were incubated in oxygenated aCSF at 32 °C for at least 1 h, then maintained at room temperature (22–25 °C) for another 30 min before recording. Dorsal hippocampal slices were gently transferred to a recording chamber (RC-27, Warner Instruments, Hamden, CT) at room temperature. The chamber was perfused with circulated oxygenated aCSF at a flow rate of 2–3 ml/min. CA1 pyramidal neurons were visually identified in slices using an infrared-differential interference contrast microscope (Scientifica, UK). Whole-cell patch-clamp recordings were performed using borosilicate glass pipettes (ID: 0.68 mm, OD: 1.2 mm, WPI, Sarasota, FL) of 3–5 MΩ pulled with a micropipette puller (P-1000; Sutter Instrument, Novato, CA). Recording pipettes were filled with internal solution containing (in mM) 115 CsMeSO₃, 20 CsCl, 10 Hepes, 0.6 EGTA, 4 MgATP, 0.3 Na₃GTP, 1 QX-314, 2.5 MgCl₂, and 10 Na₂-Phosphocreatine (pH 7.3 with CsOH, osmolarity 285 mM). mEPSCs were recorded by holding neurons at −70 mV in voltage-clamp mode without series resistance and liquid junction compensation. mEPSCs were recorded in the presence of 1 μM tetrodotoxin and 100 μM picrotoxin (PTX).

*Synaptic plasticity measurements:* Slices were incubated in oxygenated aCSF at 32 °C for at least 30 min, then maintained at room temperature for another 30 min before recording. Slices were gently transferred to the recording chamber at room temperature. Chamber was perfused with non-circulated oxygenated aCSF at a flow rate of 2–3 ml/min. fEPSPs were evoked by a concentric bipolar stimulating electrode (inner diameter: 25 μm; outer diameter: 125 μm, FHC Inc., Bowdoin, ME) connected to a constant current stimulus isolator (SYS-A365R, WPI, Sarasota, FL). Recordings were performed with low resistance (1–3 MΩ) glass pipettes filled with aCSF. fEPSPs at dorsal hippocampal Schaffer collateral-CA1 synapses were recorded. The recording electrode was placed 400 μm away from the stimulating electrode in the CA1 stratum radium area. For input–output measurement, fEPSP slope was recorded by increasing the stimulation intensity (0.1-ms pulse width) from 0 to 100 μA in a 20-μA increment. For PPR, LTP, and LTD, stimulation intensities were adjusted to give 50% of the fEPSP slope. After recording 20 min stable baseline, tetanic stimuli (HFS: 4 trains of 100 pulses at 100 Hz, spaced by 20 s) or low-frequency stimulation (LFS: 900 pulses at 1 Hz) were delivered to induce LTP or LTD, respectively. LTP and LTD were recorded for 1 h after stimuli were applied.

Experimenters were blind to genotype. Signals were acquired with Multiclamp 700B and Digidata 1550A (Molecular Devices, San Jose, CA). Data were low-pass filtered at 2.9 KHz and sampled at 10 kHz. mEPSCs and fEPSPs were analyzed with Clampfit 10.6 software. For mEPSCs, the detection threshold was set at 4 pA and 100 events were sampled per neuron. mEPSC decay kinetics was measured as the time elapsed from 10 to 90% of the peak amplitude of the response[74]. Data of LTP and LTD were expressed as averages of fEPSP slope for every 2 min of recordings,

and slopes in the last 10 min of recordings were averaged per animal. fEPSP decay kinetics was measured as the time elapsed from peak amplitude to baseline from input–output recordings at 40 µA stimulation.

**Behavioral tests**. Male WT and *Casp2* KO mice at 2–4 months of age were tested during the dark (active) phase of a 12-/12-h reversed light–dark cycle. Mice were moved to a holding room in the behavioral testing area, at least 1 h ahead of behavioral assays. The apparatus was cleaned with 1% Micro-90 between each trial. Automatic scoring was performed using the Ethovision XT video tracking system (Noldus, Netherlands). There was a break of at least 3 days between two behavioral tests.

*Open-field test*: The mouse was placed in an open-field arena (43.8 × 43.8 × 32.8 cm) for 30 min. Total distance moved, velocity and center zone duration were recorded automatically.

*Light–dark box test*: Mice were placed in the light chamber and allowed to explore for 5 min. Latency to enter the dark chamber, duration of time in the light chamber and number of crossings between chambers were automatically recorded.

*Spontaneous T-Maze alternation*: Spontaneous alternation in a T-maze was used to assess working memory in WT and *Casp2* KO mice. Briefly, mice received two successive nonrewarded trials in 1 day. On trial 1, mice were allowed to enter one of the two unfamiliar choice arms. After staying for 10 s in the chosen arm, mice were removed from the maze and placed at the start arm for the second trial. A correct choice equals exploration of the arm not previously explored.

*Rotarod test*: Mice were placed on a rotating rod (ENV-577M, Med Associates Inc.). The speed of rotation was gradually increased from 4 to 40 rpm over a 5 min period. Mice received three trials per day for 2 days, each trial was spaced at least 1 h apart. The latency to fall was measured.

*Fear conditioning test*: The mouse was placed into a Phenotyper chamber (29.2 × 29 × 30.5 cm, Noldus) equipped with an electrified floor and speaker. Training consisted of a 150 s baseline followed by three 0.75 mA foot shocks (with 30 s long 85 dB white noise tone). Conditioning testing occurred 24 h, and 1–4 weeks after the initial training. Mice were tested for contextual fear memory by returning to their training chambers for 5 min. For cued conditioning, mice were placed into a novel environment for a 3 min baseline followed by a 3 min tone presentation. Freezing behavior was automatically recorded by using the Ethovision XT video tracking system.

*Morris water maze*: Spatial learning and memory was assessed in the MWM using a video-tracking system (EthoVision XT, Noldus). Mice were tested in four trials per day in a pool (48″ in diameter, water temperature was held at 23 °C) surrounded by distal visual cues. A circular platform (10-cm diameter) was used as the goal platform. Escape latency and swim path were recorded automatically. Testing was conducted in three phases: visible platform training (day 0), acquisition training (days 1–10), and reversal acquisition training (days 11–16). Retention performance (60 s) was evaluated with the platform removed on days 11, 17, and 27. Time spent in the target quadrant, latency to the platform, platform area crossing numbers, and duration served as the dependent variables.

**Statistical analysis**. Statistical analyses were performed using SPSS. Student's *t* test, one-way or two-way ANOVA, followed by post hoc test were used as appropriate. The minimal level of significance was set at $p < 0.05$.

**Reporting summary**. Further information on research design is available in the Nature Research Reporting Summary linked to this article.

## Data availability
The data that support the findings of this study are available from the corresponding author upon reasonable request. The source data for all figures are provided with the paper.

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

## Acknowledgements

We thank Dr Sally Kornbluth and Dr Kenkyo Matsuura for kindly providing pCasp2pro-VN and pCasp2pro-VC plasmids. We thank Chien-cheng Shih for his help in performing time-lapse imaging of dendritic spines and Shaw-wen Wu and Jessica Houtz for critical comments on this paper. This work was supported by National Institutes of Health grants NS073930 (B.X.), DK103335 (B.X.), DK105954 (B.X.), and MH105482 (K. A.M.). Z.X.X. and H.X. were partially supported by a Training Grant in Alzheimer's Drug Discovery from the Lottie French Lewis Fund of the Community Foundation for Palm Beach and Martin Counties.

## Author contributions

Z.X.X. and B.X. conceived and designed this study, carried out data analysis, and wrote the paper. B.X. supervised and coordinated the project. Z.X.X. produced mice and carried out all neuronal, biochemical and behavioral experiments. J.W.T. performed electro-physiological experiments and analyzed electrophysiological data. C.J.H. assisted Z.X.X. in measurement of spine density and spine head diameter. H.X. prepared neuronal cultures used in the study. O.O. and K.A.M. were involved in some electrophysiological experiments. All authors were involved in editing the paper.

## Additional information

**Competing interests:** The authors declare no competing interests.

