## [Peer Review File · Nature Communications]

Reviewers' Comments:

Reviewer #1:

Remarks to the Author:

In the manuscript by Xu et al, the authors show that caspase-2 knockout mice have impairments in cognitive flexibility manifesting as increased anxiety and memory retention. They present data suggesting that this is regulated by caspase-2 mediated cleavage of Rictor, which inhibits Akt activation and increases GSK3beta activity, somehow resulting in the reduction of dendritic spine density. While the data on dendritic spine density and memory retention in the animal model is really interesting the novelty and robustness of the results are diminished somewhat by a prior paper that shows similar effects but only in an Alzheimer's model (see below). In addition, the mechanistic data supporting a role for caspase-2 regulating the Akt pathway is not sufficiently strong.

Major points

1. A previous paper by Pozueta et al (Nature communications 2013) studied dendritic spine alterations in an alzheimer's model. In this paper the authors show that caspase-2 loss prevents the decrease in dendritic spine density induced by Aβ. While the current manuscript claims that this is the first study to look at the role of caspase-2 in spine density and behaviours in normal mice, the Pozueta paper did include untreated mice in their studies and showed no difference in spine density. There was also no difference in memory deficits reported between WT and Casp2KO mice. While the Pozueta paper is referenced throughout the current manuscript the contradicting results and the possible reasons for this are not acknowledged or discussed.

2. The paper relies heavily on the use of a caspase-2 inhibitor VDVAD. This peptide sequence is not specific for caspase-2 and can be efficiently cleaved by other caspases including caspase-3 (McStay et al CDD 2008). While most of the experiments are backed up with shRNA data figure 2D uses the VDVAD substrate as the sole evidence to claim specific activity of caspase-2 induced by NMDA in dendrites. Without further controls to show specificity, these data are insufficient to show caspase-2 is activated.

3. In figure 2 the authors use immunostaining of a HA tagged caspase-2 expression construct to show localization of caspase-2. However, overexpression of caspase-2 is known to induce localization artifacts so this should be confirmed by staining for the endogenous protein.

4. Figure 2c claims to show that caspase-2 translocates from the nucleus to the cytoplasm with NMDA treatment. However, while the densitometry supports this, the accompanying blot is very confusing. There are no fractionation controls (actin should not be in the nuclear fraction) and the corresponding increases and decreases in caspase-2 levels are not evident by studying the western blot.

5. Following from the previous point, the reliance on densitometry is a major issue with the paper. The densitometry does not always match up with what is visible in the raw data (Figure 2c, 4c (S-GluR1), 5a, 5b, 5d) or does not have a matching western to inspect (Fig S5c). I am not convinced that the small differences that are statistically significant upon densitometry would have a biological effect. As an aside, I would strongly encourage the authors to present the immunoblots as greyscale because the contrast of the bands vs background is often difficult to see, especially with the red.

6. The data suggesting that rictor is a substrate for caspase-2 is weak. This is all based on overexpression of caspase-2 and a poor quality blot (figure 5 c). To show that rictor is indeed a substrate for caspase-2, the authors should map the cleavage site, mutate it to show that cleavage is blocked or show that cleavage is impaired in the absence of caspase-2. Given that this is a key entry point to the pathway that is claimed to be responsible for the phenotypic effects shown the evidence

that this is a caspase-2 substrate needs to be much stronger.

7. Similarly, there is no corroborating evidence that the mTOR/Akt/Gsk3b pathway is responsible for the caspase-2-dependent effects on spine density or memory. The conclusions are all based on relative protein levels but no functional assays are presented to support the involvement of this pathway.

Minor points

1. Supplementary Figure 3: The authors state that the results in this figure suggest that caspase-2 is involved in LTD-induced spine shrinkage because inhibiting caspase-2 abolished NDMA-induced spine shrinkage. However, the caspase-2 inhibited spine heads are smaller to start out. Could an alternate explanation be that they are just unable to decrease any further?

2. On page 6, the authors state that procaspase-2 is partially active and reference Baliga et al (CDD 2004). Procaspase-2 is not partially active and the referenced paper did not demonstrate this. What they showed was that when procaspase-2 undergoes dimerization it is partially active, but this dimerization requires a stimulus to induce caspase-2 recruitment to its signaling complex. Therefore, increased levels of caspase-2 cannot be used to suggest enhanced activity

3. Figure 2D y-axes are labeled as VDVAC instead of VDVAD.

4. The methods list procedures for Golgi staining but this is not used in the manuscript as far as I can tell.

Reviewer #2:

Remarks to the Author:

This interesting manuscript describes the role of Caspase-2 in regulation of dendritic spines, LTD and learning and memory. A proposed molecular mechanism is presented in which caspase-2 cleavage of Rictor deactivates mTORC2, resulting in reduced activation of Akt via pSer473 and reduced inhibitory phosphorylation of GSK3beta at pSer9, and ultimately leading to AMPAR internalization and degradation.

Overall the study is nicely conducted and presented, and the findings are an important extension of previous work on caspases (mostly caspase-3) in LTD and the normal functions of caspase-2 in behavior. However, there are a number of technical issues that should be addressed before publication can be recommended:

1. Fig. 1a-c- spine number is measured but other morphometric parameters should also be examined. Although spine size is reported in a later figure, it also looks as though the spines might be longer with caspase-2 inhibition and should be quantified. This can easily be analyzed with existing images.

2. Supplementary Fig. 1c – This figure is not well described. There is no description of caspase 2S in the methods. It is also unclear how the rescue was performed, was this a transfection of GFP + shRNA followed by an infection of Caspase-2L or -2S, or was it a triple transfection? How is it known that there is successful expression of Caspase-2L or 2S? Overexpression should be documented. How is it known that there is successful co-transfection of the shRNA with the GFP? Control experiments could be done showing that GFP cells lack caspase-2 protein (or activity using fluorescent Z-VDVAD reporter).

3. Fig. 2 – the authors suggest that nuclear Caspase-2 decreases with NMDA treatment; however, using the fluorescent VDVAD reporter, there is no decrease in the nucleus (Fig. 2d), which seems to be contradictory and should be discussed.

4. Fig 4 – It is shown that total GluA1 levels are increased in the KO mouse hippocampus (Fig. 5a), but in the hippocampal neurons (Fig. 5c) there doesn't seem to be any change in surface or total GluA1 (veh KO vs. veh WT). This should be quantified and if discrepant should be addressed. Along these lines, in Supplementary Fig. 4 – the surface GluA1 levels look less in Casp-2 KO neurons, even though the total GluA1 is reported to be higher. Surface GluA1 should be quantified along with internalized.

5. Fig. 5b – it is stated that Rictor levels are "greatly" increased in Casp2 KO mice, but the figure seems to show a barely perceptible increase of Rictor in the KO mice, certainly less than the 2-fold indicated by the graph.

6. Fig. 5d – some of the westerns are difficult to understand with Casp2 overexpression - why does mTOR level go down, pS6K go up, and total GSK3beta go down dramatically? These effects are not adequately discussed.

7. Fig. 5 – The authors suggest that the Caspase-2  rictor  mTORC2  GSK3beta pathway "should be" important for removal of AMPARs and NMDAR-LTD. This may be one of the most critical experiments but is not tested. A relatively straightforward way to test this is to use the chemical LTD paradigm. Stimulating WT hippocampal neurons with NMDA should lead to Rictor cleavage, reduction in S473 Akt, and reduced S9-GSK3beta, whereas in Casp2 KO neurons NMDA should not promote any of these things.

8. In the discussion it is mentioned that the faster decay kinetics in Casp2 KO mice may be due to a change in AMPAR subunit composition, perhaps GluA1 homomers. This can be tested by the addition of philanthotoxin-433, an inhibitor of homomeric GluA1 AMPARs.

Minor points:

1. Fig. 1c – the y-axis starts at 0.6 rather than 0, which is slightly misleading and should be avoided.

2. Arrangement of figures is sometimes confusing:

-Fig. 3a-c should go in Supplementary, and Supplementary Fig. 3a,b added to Fig. 3.

-Fig. 6a should go in Supplementary Fig. 6, while Supplementary Fig. 6o should go into Fig. 6

-Supplementary Fig. 5c should go into Fig. 5.

3. There are numerous behavioral tests; it would be helpful to have a descriptive label on the figure panels to make it more clear which test is being done.

4. Several typos are evident throughout and needs better proofreading. Examples: Fig. 2d graphs say VDVAC instead of VDVAD. Also, Fig. 4c occasionally use the old GluR nomenclature instead of GluA.

Reviewer #3:

Remarks to the Author:

This paper reports interesting results on the electrophysiological, behavioral and biochemical alterations in the brain as a function of caspase-2 level and presumably activity. Much of the work is novel and some retrads work from other groups where caspase-2 function was measured in rodent models of human disease. The most remarkable and potentially important finding is that caspase-2 is needed for cleavage of Rictor and that the mTORC2 actions are regulated by this cleavage. There are,

however, some weaknesses to the studies. The activity of Caspase 2 is measured using the pseudosubstrate reporter VDVAC-FITC which is capable of binding to caspases other than Caspase 2. A biotinylated VAD (bVAD) assay with immunodetection is far more specific but will not yield the spatial data. The key deficit is that while the data are convincing that Caspase 2 is necessary for Rictor cleavage there is no evidence that caspase 2 actually cleaves Rictor. Rictor has been reported to be cleaved by Caspase 3 and if, as the authors believe, caspase 2 is an initiator caspase for caspase 3 then it is possible cleavage is due to caspase 3 - does this still happen if caspase 3 is downregulated? More globally it is unclear whether caspase 2 is a classical initiator caspase. Papers from the Troy lab have shown that Caspase 2 can kill in absence of caspase 3 and that, depending on the circumstance, either caspase 2 or caspase 3 can cause cell death in a trophic factor deprivation model.

We sincerely thank all of the reviewers for their insightful and constructive comments. Significant revisions in the manuscript text are highlighted in blue. Our point-by-point response to the comments is listed below.

Reviewer #1

In the manuscript by Xu et al, the authors show that caspase-2 knockout mice have impairments in cognitive flexibility manifesting as increased anxiety and memory retention. They present data suggesting that this is regulated by caspase-2 mediated cleavage of Rictor, which inhibits Akt activation and increases GSK3beta activity, somehow resulting in the reduction of dendritic spine density. While the data on dendritic spine density and memory retention in the animal model is really interesting, the novelty and robustness of the results are diminished somewhat by a prior paper that shows similar effects but only in an Alzheimer's model (see below). In addition, the mechanistic data supporting a role for caspase-2 regulating the Akt pathway is not sufficiently strong.

Major points

1. A previous paper by Pozueta et al (Nature communications 2013) studied dendritic spine alterations in an alzheimer's model. In this paper the authors show that caspase-2 loss prevents the decrease in dendritic spine density induced by Aβeta. While the current manuscript claims that this is the first study to look at the role of caspase-2 in spine density and behaviours in normal mice, the Pozueta paper did include untreated mice in their studies and showed no difference in spine density. There was also no difference in memory deficits reported between WT and Casp2KO mice. While the Pozueta paper is referenced throughout the current manuscript the contradicting results and the possible reasons for this are not acknowledged or discussed.

With regard to spine density, Pozueta et al. used Dil to label dendritic spines. Dil may not get into distal dendrites efficiently. Therefore, spine density could be determined in proximal dendrites in the study by Pozueta et al. This finding would be consistent with our result. We found that *Casp2* KO mice only have a higher spine density in distal dendrites than WT mice (Supplementary Fig 1h). With regard to memory deficits, Pozueta et al. only examined learning and memory during normal training in water maze tests. We also did not detect that *Casp2* KO mice have any alteration in learning and memory in the Morris water maze test during normal training (Supplementary Fig. 1f-l). We did detect that *Casp2* KO mice have better contextual fear memory (Fig. 6d, e), worse performance during reverse training in the Morris water maze test (Fig. 6g), and better remote memory in the Morris water maze test (Fig. 6h-j). These behavioral tests were not performed in the study by Pozueta et al. We discuss the difference between the two studies in this revised manuscript (the 2nd and 3rd paragraphs on p16).

2. The paper relies heavily on the use of a caspase-2 inhibitor VDVAD. This peptide sequence is not specific for caspase-2 and can be efficiently cleaved by other caspases including caspase-3 (McStay et al CDD 2008). While most of the experiments are backed up with shRNA data figure 2D uses the VDVAD substrate as the sole evidence to claim specific activity of caspase-2 induced by NMDA in dendrites. Without further controls to show specificity, these data are insufficient to show caspase-2 is activated.

We have employed the bimolecular fluorescence complementation system to substantiate our claim that NMDA increases the caspase-2 activity in dendrites. The new result is shown in Figure 2d.

3. In figure 2 the authors use immunostaining of a HA tagged caspase-2 expression construct to

show localization of caspase-2. However, overexpression of caspase-2 is known to induce localization artifacts so this should be confirmed by staining for the endogenous protein.

We have tested several commercially available caspase-2 antibodies. Unfortunately, the best one still recognizes non-specific proteins. We include a large immunoblot in Supplementary Fig. 1a to show that caspase-2 antibodies recognize a non-specific protein in addition to caspase-2. Currently, we do not have a good tool to examine the subcellular localization of endogenous caspase-2.

4. Figure 2c claims to show that caspase-2 translocates from the nucleus to the cytoplasm with NMDA treatment. However, while the densitometry supports this, the accompanying blot is very confusing. There are no fractionation controls (actin should not be in the nuclear fraction) and the corresponding increases and decreases in caspase-2 levels are not evident by studying the western blot.

We analyzed proteins from cytosolic and nuclear fractions with histone H3 antibodies. The immunoblot shows that the nuclear fraction is rich in H3 while the cytosolic fraction is free of H3, indicating that our fractionation is reasonably good. We have included this new result in this revised manuscript (Supplementary Fig. 2b). Many studies have already shown that actin is in the nucleus (reviewed in Falahzadeh et al., 2015, Cell J. 17: 7-14). We have converted immunoblots from color to grey throughout the manuscript. It is easier to perceive a difference in greyscale.

5. Following from the previous point, the reliance on densitometry is a major issue with the paper. The densitometry does not always match up with what is visible in the raw data (Figure 2c, 4c (S-GluR1), 5a, 5b, 5d) or does not have a matching western to inspect (Fig S5c). I am not convinced that the small differences that are statistically significant upon densitometry would have a biological effect. As an aside, I would strongly encourage the authors to present the immunoblots as greyscale because the contrast of the bands vs background is often difficult to see, especially with the red.

As suggested, we have converted immunoblots from color to grey throughout the manuscript. In greyscale, there is a better match in densitometry and band appearance.

6. The data suggesting that rictor is a substrate for caspase-2 is weak. This is all based on overexpression of caspase-2 and a poor-quality blot (figure 5 c). To show that rictor is indeed a substrate for caspase-2, the authors should map the cleavage site, mutate it to show that cleavage is blocked or show that cleavage is impaired in the absence of caspase-2. Given that this is a key entry point to the pathway that is claimed to be responsible for the phenotypic effects shown the evidence that this is a caspase-2 substrate needs to be much stronger.

We have carried out additional experiments to strengthen the argument that Rictor is a substrate for caspase-2. The immunoblot in Figure 5c appears poor due to low levels of endogenous Rictor. To better detect Rictor breakdown products, we overexpressed Rictor-Myc alone or Rictor-Myc + caspase-2 in HEK293 cells. With caspase-2 overexpression, Rictor was broken down into multiple fragments (Supplementary Fig. 5b). This observation not only supports that Rictor is a direct or indirect substrate of caspase-2, but also indicates that Rictor is degraded at multiple cleavage sites in the presence of caspase-2. We did not detect multiple Rictor breakdown products in neurons (Fig. 5c), likely due to low levels of endogenous Rictor. Because of the multiple cleavage sites, it becomes extremely difficult to block Rictor degradation through mutagenesis of cleavage sites. Thus, we did not carry out experiments to

map cleavage sites. To further test if Rictor is a substrate of caspase-2, we examined whether chemical LTD induces Rictor degradation and whether the degradation requires caspase-2. We found that NMDA treatment reduced levels of Rictor in cultured WT, but not *Casp2* KO, hippocampal neurons (Fig. 5e). We think that these new data greatly strengthen the claim that Rictor is a caspase-2 substrate.

7. Similarly, there is no corroborating evidence that the mTOR/Akt/Gsk3b pathway is responsible for the caspase-2-dependent effects on spine density or memory. The conclusions are all based on relative protein levels but no functional assays are presented to support the involvement of this pathway.

We found that chemical LTD reduced levels of Rictor, active S473-Akt and inactive S9-GSK3 β in cultured WT, but not *Casp2* KO, hippocampal neurons (Fig. 5e). Furthermore, we found that Rictor knockdown decreased spine density (Fig. 5f), which is consistent with a previous report that *Rictor* knockout reduced spine density in CA1 pyramidal neurons (Huang et al., 2013, Nature Neuroscience). Interestingly, we found that Rictor overexpression increased spine density, which was blocked by caspase-2 co-overexpression (Supplementary Fig. 5d). These new data in the revised manuscript should provide additional corroborating evidence that the mTORC2/Akt/ GSK3 β pathway is responsible for the caspase-2-dependent effects on spine morphology (spine head size and spine density).

Minor points

1. Supplementary Figure 3: The authors state that the results in this figure suggest that caspase-2 is involved in LTD-induced spine shrinkage because inhibiting caspase-2 abolished NDMA-induced spine shrinkage. However, the caspase-2 inhibited spine heads are smaller to start out. Could an alternate explanation be that they are just unable to decrease any further?

NMDA treatment also induced shrinkage of small dendritic spines in control neurons (Supplementary Fig. 3a, c). Thus, we think that the alternate explanation is unlikely.

2. On page 6, the authors state that procaspase-2 is partially active and reference Baliga et al (CDD 2004). Procaspase-2 is not partially active and the referenced paper did not demonstrate this. What they showed was that when procaspase-2 undergoes dimerization it is partially active, but this dimerization requires a stimulus to induce caspase-2 recruitment to its signaling complex. Therefore, increased levels of caspase-2 cannot be used to suggest enhanced activity.

We have corrected the error.

3. Figure 2D y-axes are labeled as VDVA instead of VDVA.

Corrected.

4. The methods list procedures for Golgi staining but this is not used in the manuscript as far as I can tell.

Golgi staining was used for the data shown in Fig. 1c.

Reviewer #2

This interesting manuscript describes the role of Caspase-2 in regulation of dendritic spines, LTD and learning and memory. A proposed molecular mechanism is presented in which caspase-2 cleavage of Rictor deactivates mTORC2, resulting in reduced activation of Akt via pSer473 and reduced inhibitory phosphorylation of GSK3beta at pSer9, and ultimately leading to AMPAR internalization and degradation.

Overall the study is nicely conducted and presented, and the findings are an important extension of previous work on caspases (mostly caspase-3) in LTD and the normal functions of caspase-2 in behavior. However, there are a number of technical issues that should be addressed before publication can be recommended:

1. Fig. 1a-c- spine number is measured but other morphometric parameters should also be examined. Although spine size is reported in a later figure, it also looks as though the spines might be longer with caspase-2 inhibition and should be quantified. This can easily be analyzed with existing images.

We have quantified spine head size and spine length. The new data are shown in Supplementary Fig. 1c-f.

2. Supplementary Fig. 1c – This figure is not well described. There is no description of caspase 2S in the methods. It is also unclear how the rescue was performed, was this a transfection of GFP + shRNA followed by an infection of Caspase-2L or -2S, or was it a triple transfection? How is it known that there is successful expression of Caspase-2L or 2S? Overexpression should be documented. How is it known that there is successful co-transfection of the shRNA with the GFP? Control experiments could be done showing that GFP cells lack caspase-2 protein (or activity using fluorescent Z-VDVAD reporter).

The figure becomes Supplementary Fig. 1g in this revised manuscript. We have added one paragraph in the Methods section to describe constructs for expressing shRNA-resistant caspase-2L and -2S and the rescue experiment. Three constructs (GFP-actin construct, shRNA-expressing construct and Myc-tagged caspase-2 expressing construct) were co-transfected into rat hippocampal neurons on DIV14. To test the co-transfection efficiency under our experimental conditions, we transfected three constructs (expressing GFP, an shRNA + tdTomato, and caspase-2L-Myc, respectively) into cultured neurons. We found that the three constructs could get into transfected neurons together in nearly all cases. One example is shown below. In Supplementary Fig. 1b, we show that *Casp2* shRNA is able to knock down caspase-2.

3. Fig. 2 – the authors suggest that nuclear Caspase-2 decreases with NMDA treatment; however, using the fluorescent VDVAD reporter, there is no decrease in the nucleus (Fig. 2d), which seems to be contradictory and should be discussed.

The fluorescent VDVAD reporter is used to indicate the level of activated caspase-2. Although NMDA treatment decreases the amount of nuclear caspase-2, it may not significantly alter the amount of activated caspase-2 in the nucleus.

4. Fig 4 – It is shown that total GluA1 levels are increased in the KO mouse hippocampus (Fig. 5a), but in the hippocampal neurons (Fig. 5c) there doesn't seem to be any change in surface or total GluA1 (veh KO vs. veh WT). This should be quantified and if discrepant should be addressed. Along these lines, in Supplementary Fig. 4 – the surface GluA1 levels look less in Casp-2 KO neurons, even though the total GluA1 is reported to be higher. Surface GluA1 should be quantified along with internalized.

We have compared levels of GluA1, Rictor, S473-Akt and S9-GSK3 β in untreated WT and KO neurons in culture. The new data are shown in Figure 5c. We did not detect a significant difference. We have discussed the discrepancy between in vivo and in vitro data (the 2nd paragraph on p11). We have quantified surface GluA1 and internalized GluA1 (Supplementary Fig. 4).

5. Fig. 5b – it is stated that Rictor levels are “greatly” increased in Casp2 KO mice, but the figure seems to show a barely perceptible increase of Rictor in the KO mice, certainly less than the 2-fold indicated by the graph.

We now show the immunoblot in greyscale. It is easier to perceive a difference in greyscale than in color. We have deleted the word “greatly”.

6. Fig. 5d – some of the westerns are difficult to understand with Casp2 overexpression - why does mTOR level go down, pS6K go up, and total GSK3beta go down dramatically? These effects are not adequately discussed.

We have increased the discussion on the protein level changes in this revised manuscript.

7. Fig. 5 – The authors suggest that the Caspase-2  rictor  mTORC2  GSK3beta pathway “should be” important for removal of AMPARs and NMDAR-LTD. This may be one of the most critical experiments but is not tested. A relatively straightforward way to test this is to use the chemical LTD paradigm. Stimulating WT hippocampal neurons with NMDA should lead to Rictor cleavage, reduction in S473 Akt, and reduced S9-GSK3beta, whereas in Casp2 KO neurons NMDA should not promote any of these things.

We have done the suggested experiment. We found that NMDA treatment significantly reduced levels of Rictor, S473-Akt and S9-GSK3 β in cultured WT, but not *Casp2* KO, hippocampal neurons (Fig. 5e). Because the endogenous level of Rictor in neurons is low, we could not detect Rictor cleavage products after the NMDA treatment.

8. In the discussion it is mentioned that the faster decay kinetics in Casp2 KO mice may be due to a change in AMPAR subunit composition, perhaps GluA1 homomers. This can be tested by the addition of philantotoxin-433, an inhibitor of homomeric GluA1 AMPARs.

Philantotoxin-433 is not commercially available anymore. Instead, we used Naspm to block GluA2-lacking AMPARs (including homomeric GluA1 AMPARs). Naspm addition abolished the difference in decay kinetics between WT and KO mice, supporting that the faster decay kinetics in Casp2 KO mice may be due to an increase in the fraction of homomeric GluA1 AMPARs. The new result is shown in Figure 4c.

Minor points:

1. Fig. 1c – the y-axis starts at 0.6 rather than 0, which is slightly misleading and should be avoided.

Corrected.

2. Arrangement of figures is sometimes confusing:

-Fig. 3a-c should go in Supplementary, and Supplementary Fig. 3a,b added to Fig. 3.

-Fig. 6a should go in Supplementary Fig. 6, while Supplementary Fig. 6o should go into Fig. 6

-Supplementary Fig. 5c should go into Fig. 5.

We have revised the figures according to the suggestions.

3. There are numerous behavioral tests; it would be helpful to have a descriptive label on the figure panels to make it more clear which test is being done.

We have added the names of some behavioral tests to figure panels.

4. Several typos are evident throughout and needs better proofreading. Examples: Fig. 2d graphs say VDVAC instead of VDVA. Also, Fig. 4c occasionally use the old GluR nomenclature instead of GluA.

We have corrected the errors.

Reviewer #3

This paper reports interesting results on the electrophysiological, behavioral and biochemical alterations in the brain as a function of caspase-2 level and presumably activity. Much of the work is novel and some retreads work from other groups where caspase-2 function was measured in rodent models of human disease. The most remarkable and potentially important finding is that caspase-2 is needed for cleavage of Rictor and that the mTORC2 actions are regulated by this cleavage.

There are, however, some weaknesses to the studies. The activity of Caspase 2 is measured using the pseudosubstrate reporter VDVAC-FITC which is capable of binding to caspases other than Caspase 2. A biotinylated VAD (bVAD) assay with immunodetection is far more specific but will not yield the spatial data.

We have employed the bimolecular fluorescence complementation system to substantiate our claim that NMDA increases the caspase-2 activity in dendrites. The assay is based on the finding that dimerization activates caspase-2. The new result is shown in Figure 2d.

The key deficit is that while the data are convincing that Caspase 2 is necessary for Rictor cleavage there is no evidence that caspase 2 actually cleaves Rictor. Rictor has been reported to be cleaved by Caspase 3 and if, as the authors believe, caspase 2 is an initiator caspase for

caspase 3 then it is possible cleavage is due to caspase 3 - does this still happen if caspase 3 is downregulated? More globally it is unclear whether caspase 2 is a classical initiator caspase. Papers from the Troy lab have shown that Caspase 2 can kill in absence of caspase 3 and that, depending on the circumstance, either caspase 2 or caspase 3 can cause cell death in a trophic factor deprivation model.

We have carried out additional experiments to substantiate the claim that Rictor is a substrate for caspase-2. To better detect Rictor breakdown products, we overexpressed Rictor-Myc alone or Rictor-Myc + caspase-2 in HEK293 cells. With caspase-2 overexpression, Rictor was broken down into multiple fragments (Supplementary Fig. 5b). This observation not only supports that Rictor is a substrate of caspase-2, but also indicates that Rictor is degraded at multiple cleavage sites in the presence of caspase-2. We did not detect multiple Rictor breakdown products in neurons (Fig. 5c), likely due to low levels of endogenous Rictor. Additionally, we examined whether chemical LTD induces Rictor degradation and whether the degradation requires caspase-2. We found that NMDA treatment reduced levels of Rictor in cultured WT, but not *Casp2* KO, hippocampal neurons (Fig. 5e).

As pointed out by the reviewer, we do not know if caspase-2 cleaves Rictor directly or through other proteases.

Reviewers' Comments:

Reviewer #1:

Remarks to the Author:

In the revised manuscript by Xu et al the authors added a some additional experiments, text and quantitation to strengthen their original manuscript that proposes that caspase-2 reduces of dendritic spine density through cleavage of Rictor to inhibits Akt activation and increased GSK3beta activity. This mechanism is proposed to be responsible for the impairments in cognitive flexibility observed in caspase-2 knockout mice.

The authors provided a response to all my previous comments and the manuscript is somewhat strengthened by the responses to all the reviewers. The data that caspase-2 has an effect on spine density and cognitive abilities in mice is reasonably strong and novel and they added a nice BiFC-based experiment as an alternative means to measure caspase-2 activation. Some of their responses were a little lacking in substance. For example, for figure 2a, I acknowledge that finding caspase-2 antibodies for immunostaining with sufficient specificity is difficult however this does not negate the concern that overexpression of caspase-2 may result in localization artifacts, which they did not address. This is a relatively minor concern.

The major issue that persists with this paper is that the data added to strengthen the claim that rictor is a substrate for caspase-2 (or indeed THE substrate that mediates all the effects) is still not very convincing. The authors add a few new experiments for this:

1. They show that co-expression of caspase-2 and rictor results in multiple smaller bands and claim that this implies a number of cleavage sites which would be too difficult to identify. While this would be time consuming and possibly outside the scope of this paper, I think the authors are overstating this somewhat. A very quick scan of the murine rictor protein sequence (NP_084444) shows that there are only 4 consensus caspase cleavage DXXD motifs which would give N-terminal fragments of predicted sizes of 18kD, 30 kD, 100 kd and 148 kd. On the Myc blot (S5c) there are only two bands that would match up with these sizes (30kD and 100kD) providing two candidate sites to mutate.
2. They show that the levels of Rictor decrease with NMDA treatment in caspase-2 WT neurons but not in knockouts (5E). However, the decrease is marginal at best. In addition there is a large variance in the levels of rictor in both the vehicle and NMDA treated groups in the knockout cohorts.
3. They show that silencing of rictor reduces spine density and conversely overexpression of rictor increases spine density that is in turn inhibited by caspase-2 overexpression. This is the strongest of the new data added and provides some functional, if somewhat correlative, links between the two proteins.

Overall, I am not entirely convinced by the mechanism they present or that rictor is a caspase-2 substrate, but I think the authors have provided some evidence for a functional relationship between the two proteins. Taking the study as a whole, there is possibly enough novelty and interesting possible new functions for caspase-2 that it is worth publishing.

Additional points:

1. In figure 3D. There is considerable fluorescence in the vehicle control that is not addressed. Does this mean that caspase-2 has a baseline level of activation in these cells or is this non-specific background (quite common with this technique)
2. A number of the blot panels have undergone a vertical inversion since the previous submission (Fig

2b, S1a, S2a, 5d). This doesn't appear to change any results and I assume this was just a mistake that was corrected but it was a little odd that it was so prevalent.

3. Figure 2b has had an extra actin panel added and the PSD95 and Syn blots are new. It is unclear why the two actins are there and with which panels they match up.

4. Line 137: Typo – dimerization should read dimerized

5. Line 271: Typo – Based on the (the word on is missing)

Reviewer #2:

Remarks to the Author:

The authors have addressed the critiques adequately. I recommend that the manuscript be accepted for publication.

Reviewer #3:

None

Point-by-point responses to comments

We thank the reviewers for their careful insightful comments and careful reading of the manuscript.

Reviewer #1 (Remarks to the Author):

In the revised manuscript by Xu et al the authors added some additional experiments, text and quantitation to strengthen their original manuscript that proposes that caspase-2 reduces of dendritic spine density through cleavage of Rictor to inhibits Akt activation and increased GSK3beta activity. This mechanism is proposed to be responsible for the impairments in cognitive flexibility observed in caspase-2 knockout mice.

The authors provided a response to all my previous comments and the manuscript is somewhat strengthened by the responses to all the reviewers. The data that caspase-2 has an effect on spine density and cognitive abilities in mice is reasonably strong and novel and they added a nice BiFC-based experiment as an alternative means to measure caspase-2 activation. Some of their responses were a little lacking in substance. For example, for figure 2a, I acknowledge that finding caspase-2 antibodies for immunostaining with sufficient specificity is difficult however this does not negate the concern that overexpression of caspase-2 may result in localization artifacts, which they did not address. This is a relatively minor concern.

The major issue that persists with this paper is that the data added to strengthen the claim that rictor is a substrate for caspase-2 (or indeed THE substrate that mediates all the effects) is still not very convincing. The authors add a few new experiments for this:

1. They show that co-expression of caspase-2 and rictor results in multiple smaller bands and claim that this implies a number of cleavage sites which would be too difficult to identify. While this would be time consuming and possibly outside the scope of this paper, I think the authors are overstating this somewhat. A very quick scan of the murine rictor protein sequence (NP_084444) shows that there are only 4 consensus caspase cleavage DXXD motifs which would give N-terminal fragments of predicted sizes of 18kD, 30 kD, 100 kd and 148 kd. On the Myc blot (S5c) there are only two bands that would match up with these sizes (30kD and 100kD) providing two candidate sites to mutate.

There are 5 DXXD sites in the murine Rictor protein. In addition, there are other putative cleavage sites for caspase-2 (Kim et al., 2018, Cell 175, 133-145). We agree with the reviewer on the comment that demonstrating Rictor as a caspase-2 substrate is “outside the scope of this paper”. We point out the limitation of our study by stating that “This argument could be strengthened by mutagenesis studies validating that Rictor is a substrate for caspase-2” in the first paragraph of Discussion.

2. They show that the levels of Rictor decrease with NMDA treatment in caspase-2 WT neurons but not in knockouts (5E). However, the decrease is marginal at best. In addition, there is a large variance in the levels of rictor in both the vehicle and NMDA treated groups in the knockout cohorts.

We believe that the size of the change in the level of Rictor after NMDA treatment is physiological, given that NMDA treatment only has a small effect on dendritic spines (Fig. 3a, b). As shown in Fig. 5f, a large reduction in the level of Rictor will drastically decrease spine density.

3. They show that silencing of rictor reduces spine density and conversely overexpression of rictor increases spine density that is in turn inhibited by caspase-2 overexpression. This is the strongest of the new data added and provides some functional, if somewhat correlative, links between the two proteins.

Thanks for the complimentary comment on the new data.

Overall, I am not entirely convinced by the mechanism they present or that rictor is a caspase-2 substrate, but I

think the authors have provided some evidence for a functional relationship between the two proteins. Taking the study as a whole, there is possibly enough novelty and interesting possible new functions for caspase-2 that it is worth publishing.

We greatly appreciate this positive overall assessment of the manuscript.

Additional points:

1. In figure 3D. There is considerable fluorescence in the vehicle control that is not addressed. Does this mean that caspase-2 has a baseline level of activation in these cells or is this non-specific background (quite common with this technique)

We believe that the reviewer meant Figure 2d. A previous study shows no detectable BiFC signal in untreated HeLa cells (Bouchier-Hayes et al., 2009, Mol Cell 35, 830-840), indicating that the BiFC assay does not produce high non-specific background. The BiFC signal in vehicle-treated neurons suggests that neurons may contain active caspase-2 in some specific cytoplasmic compartments under the basal condition. We discuss this possibility in the Discussion section.

2. A number of the blot panels have undergone a vertical inversion since the previous submission (Fig 2b, S1a, S2a, 5d). This doesn't appear to change any results and I assume this was just a mistake that was corrected but it was a little odd that it was so prevalent.

We corrected the mistake.

3. Figure 2b has had an extra actin panel added and the PSD95 and Syn blots are new. It is unclear why the two actins are there and with which panels they match up.

There are two blots, one for Caspase-2 and the other for PSD95 and Syn. Thus, there are two actin blots as internal loading control.

4. Line 137: Typo – dimerization should read dimerized

Corrected.

5. Line 271: Typo – Based on the (the word on is missing)

Corrected.

Reviewer #2 (Remarks to the Author):

The authors have addressed the critiques adequately. I recommend that the manuscript be accepted for publication.

We greatly thank for the comment.